

# A tandem approach for collocated in-situ measurements of microphysical and radiative cirrus properties

Marcus Klingebiel[1,2], André Ehrlich[3], Fanny Finger[3], Timo Röschenthaler[4,5],
Suad Jakirlić[5], Matthias Voigt[6], Stefan Müller[4,6], Rolf Maser[4],
Manfred Wendisch[3], Peter Hoor[6], Peter Spichtinger[6], and Stephan Borrmann[2,6]

[1]Max Planck Institute for Meteorology, Atmosphere in the Earth System Department, Hamburg, Germany
[2]Max Planck Institute for Chemistry, Particle Chemistry Department, Mainz, Germany
[3]Leipzig Institute for Meteorology (LIM), University of Leipzig, Leipzig, Germany
[4]Enviscope GmbH, Frankfurt, Germany
[5]Institute for Fluid Mechanics and Aerodynamics, Darmstadt University of Technology, Darmstadt, Germany
[6]Institute for Atmospheric Physics, Johannes Gutenberg University Mainz, Mainz, Germany

*Correspondence to:* S. Borrmann (stephan.borrmann@mpic.de)

**Abstract.** Microphysical and radiation measurements were collected with the unique AIRcraft TOwed Sensor Shuttle (AIRTOSS) - Learjet tandem platform. It is a combination of a Learjet 35A research aircraft and an instrumented aerodynamic body, which can be detached from and retracted back to the aircraft during flight. Both platforms are equipped with radiative, cloud microphysical, trace gas (CO, $N_2O$, $O_3$ and $H_2O$) and meteorological instruments to study the inhomogeneity of cirrus as well as other layer clouds. Sophisticated numerical flow simulations were conducted in advance in order to optimally integrate a Cloud Combination Probe (CCP) inside the AIRTOSS. The tandem platform was used for the first time at altitudes up to 36 000 ft (10 970 m) during the AIRTOSS - Inhomogeneous Cirrus Experiment (AIRTOSS-ICE). AIRTOSS is connected to the aircraft by a steel wire with a length of 4000 m. Ten flights were performed above the North Sea and Baltic Sea to probe frontal cirrus, in-situ formed cirrus, and anvil outflow cirrus. The cirrus microphysical and radiative properties displayed significant inhomogeneities resolved by both measurement platforms. Data collected with the CCP show that the maximum of the observed particle number size distributions shifts with increasing altitude from 300 µm to 30 µm, which is typical for frontal, midlatitude cirrus. Theoretical considerations imply that cloud particle aggregation inside the studied cirrus is very unlikely. Consequently, diffusional growth was identified to be the dominant microphysical process. Measurements of solar downward irradiance at 670 nm wavelength on the Learjet and the sensor shuttle indicate a significant horizontal heterogeneity of the observed thin cirrus. Making use of the collocated irradiance measurements of the tandem platform, vertically resolved solar heating rates were derived. They varied by up to 6 K day$^{-1}$ in and above the cirrus layer.



## 1 Introduction

Cirrus clouds consist of ice particles and occur in the upper troposphere and lower stratosphere at temperatures below $-38\,°C$ (Boucher et al., 2014; Koop et al., 2000). The wide range of microphysical and macrophysical properties of cirrus affects the solar and terrestrial radiative budget of the

Earth-Atmosphere system. Depending on the microphysical properties, cirrus either warms or cools the layer below the clouds (Lynch, 2002; Zhang et al., 1999). Especially the ice particle shape was found by several studies, e.g., Wendisch et al. (2005), Wendisch et al. (2007), Eichler et al. (2009) or Finger et al. (2016) to determine the cirrus radiative properties such as top of atmosphere albedo or spectral radiative layer properties. Such effects of ice particle shape and surface roughness may

cause significant biases in cirrus retrievals from satellite instruments. However, most of these studies apply sensitivity studies for different ice crystal shapes using measurement-based radiative transfer simulations. These simulations do not directly link in-situ observations of ice crystal shape and cirrus radiative properties.

To better quantify the dependence of the cloud radiative forcing and cloud properties, spatially sepa-

rated observations of the cirrus microphysical and radiative properties are needed. This can be realized by consecutive measurements by one single measurement platform or collocated observations by two platforms. The first approach is limited by the (usually too large) temporal delay separating the single observation in, below, and above the cloud. Collocated measurements using two coordinated aircraft were attempted for example during the Cirrus Regional Study of Tropical Anvils and

Cirrus Layers - Florida Area Cirrus Experiment (CRYSTAL-FACE) in 2002 (Jensen et al., 2004), the Tropical Composition, Cloud and Climate Coupling (TC4) mission in 2007 (Toon, 2007) and the Radiation-Aerosol-Cloud Experiment in the Arctic Circle (RACEPAC) in 2014 (Ehrlich and Wendisch, 2015). However, as pointed out by Frey et al. (2009) and others, such arrangements with two different aircraft are subject to a number of limitations, flight safety being the largest. To min-

imize these problems, towed measurement systems have been applied for cloud research. During the CARRIBA (Cloud, Aerosol, Radiation and tuRbulence in the trade wInd regime over BArbados) project (Siebert et al., 2013) two helicopter borne platforms were applied to obtain collocated measurements of thermodynamic, turbulent, microphysical, and radiative properties within clouds. Werner et al. (2014) showed that such observations can be used to link cloud microphysical and

radiative properties and estimate the Twomey effect in shallow cumulus. However, these helicopter measurements are limited to altitudes below 3000 m and are not suited for investigating cirrus.

Frey et al. (2009) introduced a new tandem measurement platform consisting of a Learjet 35A research aircraft and an AIRcraft TOwed Sensor Shuttle (AIRTOSS), which can operate in higher altitudes and speeds ($\sim 700\,km\,h^{-1}$). AIRTOSS is a sensor pod that is attached under the right wing

of the Learjet. When the Learjet reaches the measurement area, AIRTOSS is released and towed by the aircraft. In Frey et al. (2009), AIRTOSS was only equipped with a Cloud Imaging Probe (CIP) to measure the microphysical properties of the clouds and two navigation systems to collect infor-



mation about the attitude angles and the position of the AIRTOSS. At his time, the configuration
of the tandem platform was certified only to fly up to an altitude of 25 000 ft (7620 m), which is
below the altitude where most cirrus typically occurs. A proof-of-concept campaign was conducted
in 2007 to assess on the technical feasibility, the flight safety, and to evaluate, if the performance of
the AIRTOSS is good enough for meaningful measurements of cloud microphysics and radiation.
Frey et al. (2009) show that turbulence as well as acceleration and deceleration maneuvers should
be avoided to keep roll and pitch angles in a range of $\pm\,3\,°$ that is tolerable for reliable, irradiance
measurements (by definition related to a strictly horizontal receiving plane). Combined with care-
ful data filtering (i.e. to exclude turns) the effect of horizontal misalignment of the AIRTOSS can
be minimized. Under these constraints it was found that it is possible to perform useful irradiance
measurements on the AIRTOSS platform (Frey et al., 2009). Motivated by these promising results,
an advanced AIRTOSS platform including radiative and cloud microphysical instruments was de-
veloped and certified (between 2011 and 2013) for higher altitudes up to 41 000 ft (12 500 m).

This paper focuses on the technical details of the redesigned and advanced AIRTOSS version that is
presented in Section 2. Section 3 shows the first results of collocated measurements in cirrus clouds
with the Learjet 35A and the further developed AIRTOSS platform. Two examples of how the col-
located observations can be analyzed are discussed in Section 4. Section 5 summarizes the outcome
and gives an overview of the strengths and weaknesses of the improved AIRTOSS-Learjet tandem
platform.

## 2 Technical development and properties of the AIRTOSS-Learjet tandem platform

The advanced AIRTOSS-Learjet tandem platform includes radiation sensors and an extended probe
for cloud microphysical measurements. This setup (see Figure 1) was used during the AIRTOSS -
Inhomogeneous Cirrus Experiment (AIRTOSS-ICE) in spring and autumn 2013 above the North Sea
and Baltic Sea.

Ten flights, five in spring (06.05.2013 – 08.05.2013) and five in autumn (29.08.2013 – 05.09.2013),
were performed during the AIRTOSS-ICE campaign. The release of the towed sensor shuttle was
only possible under strict safety regulations, and for this reason the measurement flights were only
performed in restricted military areas. In order to reach cirrus altitudes a full formal aeronautical
and aircraft certification had to be completed. After this complex procedure the tandem platform
consisting of the Learjet 35A and the AIRTOSS was allowed to operate at altitudes up to 41 000 ft
(12 500 m).

### 2.1 The Learjet 35A research aircraft

The aircraft of the tandem platform (see Figure 1a) is a Learjet 35A. It can reach a maximum flight
distance of 1700 NM and a maximum altitude of 45 000 ft and typically cruises at speeds between




$600\,\mathrm{km\,h^{-1}}$ and $800\,\mathrm{km\,h^{-1}}$. For scientific projects, the aircraft is equipped with a sensor pod under the left wing (see Figure 1b) and a winch for the AIRTOSS under the right wing. This additional freight limits the maximum altitude (to ~ 36 000 ft, 10 970 m) and distance. Radiative, meteorological and microphysical instruments were mounted inside the AIRTOSS as well as on the fuselage of the Learjet and are introduced in the following sections.

## 2.2 AIRcraft TOwed Sensor Shuttle (AIRTOSS)

The original body structure of the AIRTOSS belongs to the shuttle case of the type DO-SK6 and is manufactured by the *European Aeronautic Defence and Space Company* (EADS). It is used as a flight target for military training. The original case and the inner frame structure was modified for implementing scientific instruments to perform measurements for atmospheric science.

### 2.2.1 Structure of the AIRTOSS

A perspective view of the structure of the AIRTOSS is shown in Figure 2a. The internal frame consists of high-strength aviation aluminium EN AW-7075 and is separated into three sections. Structural elements on the internal frame allow all sensors to be mounted inside the AIRTOSS, which has a length of 2.89 m and a diameter of 0.24 m. The middle section includes the eyelet, which connects the AIRTOSS to the Learjet by a steel wire without electrical leads. A Cloud Combination Probe (CCP) is located in the front section, and the rear part of the sensor shuttle contains mainly the radiation instruments. The original version used the external body cover (made of glass-fibre reinforced plastic) as a mounting point for additional payload. For the modified version, the body cover is only used as covering which does not need a detailed strength calculation and certification.

The photograph in Figure 2b was taken from an accompanying second aircraft, during a test flight for the airworthiness certification procedure. Air brakes with different resistance coefficients were mounted at the winglets to compensate for the shape of the CCP and to keep the released AIRTOSS in a horizontal flight position.

During transfer flights into the measurement areas, the unreleased AIRTOSS stayed locked to the winch and was tilted such that it was closely held underneath the wing to ensure a save distance between sensors and ground during the take-off and landing maneuvers of the aircraft. The maximum length of the steel wire between the winch and AIRTOSS is 4000 m. During the AIRTOSS-ICE campaign the steel wire was only released to a length up to 914 m (3000 ft). Under these conditions and with an airspeed of $165\,\mathrm{m\,s^{-1}}$, AIRTOSS stayed approximately 180 m below and 900 m behind the Learjet. This horizontal displacement introduces a delay of about 5 s between Learjet and AIRTOSS instantaneous location. During turns also a lateral displacement is introduced. This data was rejected from the collocated analysis presented here. The tare weight of the AIRTOSS case without instruments is 27.0 kg. After including the instruments and the accessories, the total weight is 61.2 kg. To



get the position of the center of gravity, a trim weight of 1.4 kg was added in the rear section, resulting in a total weight of 62.6 kg. This is still under the maximum permitted total weight of 70 kg. Table 1 gives an overview of the masses of the included instruments and accessories.

**2.2.2 Energy consumption of the instruments**

A rechargeable battery serves as the power source for the instruments mounted inside AIRTOSS and is located in the center of gravity in the middle section. AIRTOSS reaches a continuous in air operation time of two hours. Safety regulations only permit to power the instrumentation when the AIRTOSS is detached from the Learjet. The consequence of this constraint is that the instruments

must start to operate autonomously in an ambient temperature between -30 °C and -50 °C. A suitable rechargeable battery for these circumstances is the Smart VHF Modul 20S2P (24 V, 30 Ah) from *SAFT batteries*. To save some energy, several heaters of the CCP instrument were deactivated. Only those from the CCP-CDP instrument (see Section 2.3) were running during the measurement flights. With these settings, all listed instruments in Table 1 consumed 213 W by an Voltage of 28 V.

The rechargeable battery delivers 720 Wh, which leads to an operating time of 3.5 h. However, considering that the CCP instrument turns off below a voltage of 22.6 V, in order to protect the lasers, the true operating time of the AIRTOSS is 2.5 h.

**2.3 Instrumentation for microphysical cloud particle measurements**

Different in-situ instruments were installed on board of AIRTOSS and the Learjet sensor pod during

the AIRTOSS-ICE campaign to collect information about the microphysical properties of cirrus clouds. The Cloud Combination Probe (CCP) instrument contained in the AIRTOSS is a modified version of the instrument initially manufactured by Droplet Measurement Technologies (DMT, Boulder, CO, USA). The position at the tip of the AIRTOSS assures that the instrument is not influenced by proximity of aircraft structures, wings and fuselage, which sometimes cause issues when mounted

at regular research aircraft (Weigel et al., 2016). To cover particles in a size range between 2 µm and 960 µm, the CCP contains a Cloud Imaging Probe grayscale (CCP-CIPg) and a Cloud Droplet Probe (CCP-CDP). Shattering artifacts (Jensen et al., 2009; Korolev et al., 2010) are minimized by using specially designed tips (Korolev et al., 2013) that are mounted to both instruments. Related artifacts can be identified and excluded by recorded particle-by-particle data (Field et al., 2003, 2006; de Reus

et al., 2009).

The CCP-CIPg records two dimensional shadow images in a size range between 15 µm and 960 µm with a resolution of 15 µm. Computer software, including special algorithms, is used afterwards to estimate cloud particle parameters like size, concentration, and shape (Korolev, 2007a).

In comparison to the CCP-CIPg instrument, the CCP-CDP detects particles in a smaller particle di-

ameter size range between 2 µm and 50 µm. The instrument is based on forward light-scattering with a light collection angle from 4 ° up to 12 ° and uses a laser diode with a wavelength of 658 nm. A



sample area of $0.27 \pm 0.025 \, \text{mm}^2$ was estimated by using a piezoelectric droplet generator labora-
tory setup, similar to the design of Lance et al. (2010) and Wendisch et al. (1996). The accuracy and
prior measurements of the CCP-CDP instrument are shown in Molleker et al. (2014) and Klingebiel
et al. (2015).

The Learjet was equipped with a Forward Scattering Spectrometer Probe (FSSP) inside the sensor
pod (Figure 1b). This instrument was developed by Knollenberg (1976) to measure particles in a size
range between $2 \, \mu\text{m}$ and $47 \, \mu\text{m}$ diameter and is a predecessor of the CCP-CDP. Because the FSSP
has neither mounted tips nor the feasibility to exclude shattered particles by software algorithms,
here it was mainly used for testing purposes and as a cloud indicator during the campaign. In the
future it will be replaced with more advanced instrumentation. Further details of the instrumentation
are given in Brenguier et al. (2013).

### 2.4 Spectral solar radiation measurements

To measure the up- and downward irradiance of a cirrus layer located between the Learjet and the
AIRTOSS, both platforms were equipped with the Spectral Modular Airborne Radiation measure-
ment sysTem (SMART). For each radiation component (upward/downward irradiance), SMART
combines two Zeiss Spectrometers each connected by fibre wires to an optical inlet mounted on
the top or at bottom of the AIRTOSS and the Learjet. The spectral range of SMART is between
$300 \, \text{nm}$ and $2200 \, \text{nm}$ with a resolution of $3 \, \text{nm}$ for wavelength below $1000 \, \text{nm}$ and $9 - 16 \, \text{nm}$ above
(Wendisch et al., 2001; Bierwirth et al., 2009). The upward looking radiation sensor on the Learjet
was placed on a stabilized platform to keep it horizontally aligned during the flights.

Due to the limited space inside AIRTOSS (see Figure 2a), an active horizontally stabilization of the
radiation sensors could not be realized. For this reason an Inertial Navigation System (INS) in com-
bination with a Global Positioning System (GPS) was used to record attitude and alignment angles.
This data was screened afterwards to identify and remove sections where reliable measurements
were not possible. A detailed analysis of the solar radiation instruments, the measurements in cirrus
and the scientific results of the AIRTOSS-ICE campaign are given in Finger et al. (2016).

### 2.5 Flow simulations

With the incorporation of the CDP component of the CCP the AIRTOSS overall geometry has been
altered in comparison with the design shown by Frey et al. (2009). Since the CDP is axially non-
symmetric the aerodynamic properties of the AIRTOSS were correspondingly modified with largely
unknown effects on alignment, attitude, and behavior during flight. Figure 3a shows a front view
of the AIRTOSS, which demonstrates the asymmetry introduced by the CDP. To investigate these
effects aiming at their compensation and to ensure stable flight conditions, such that radiation mea-
surements can be reliably conducted, detailed fluid flow simulations of the AIRTOSS aerodynamics
have been performed (Röschenthaler, 2013) by employing Computational Fluid Dynamics (CFD)



methodology. We recall that for the formal airworthiness directives certification of the AIRTOSS the corresponding simulations resulting in evolution of the forces and drag coefficients were mandatory. The 3D calculations were performed using the AVL-FIRE Thermo-Fluid Simulation Software (by AVL-List GmbH, Graz, Austria (AVL-Fire, 2013)) employing a finite volume discretization method based on the integral form of the general conservation law applied to polyhedral control volumes. The turbulence model adopted is a four-equation, eddy-viscosity-based turbulence model denoted by $k - \varepsilon - \zeta - f$ (Hanjalić et al., 2004). Application of the 'concept of elliptic relaxation' allows for particular attention to the flow effects close to the walls when approaching the AIRTOSS surface. In addition to the equations governing the kinetic energy of turbulence $k$ and its dissipation rate $\varepsilon$ it solves transport equations for the quantity $\zeta$, representing the ratio $\overline{\nu^2}/k$, and elliptic function $f$, with $\overline{\nu^2}$ denoting the scalar variable which behaves as the normal-to-the-wall Reynolds stress component by approaching the solid wall. Here, the $\zeta-$ quantity represents a key parameter, as it models the near-wall anisotropy influence on the relevant velocity scale in the corresponding formulation for the turbulent viscosity. The so-called 'compound wall functions' blending between the integration up to the wall with the standard equilibrium wall functions were applied for the wall treatment. They are especially advantageous for the high Reynolds-number flows enabling well-defined boundary conditions irrespective of the position of the wall-closest computational node. The numerical grid discretizing the object surface and its surrounding consists of 12.7 million cells; this grid represents appropriate refinement of a coarser grid comprising 6.9 million cells. The so-called MINMOD bounded scheme combining the 2$^{nd}$ order accurate schemes CDS (Central Differencing Scheme) and LUDS (Linear Upwind Differencing Scheme) is utilized for the discretization of the convective transport and the conventional CDS scheme for the diffusive transport.

As result detailed flow velocity fields were obtained, as well as the fields of turbulence quantities, drag coefficients and aerodynamic forces. The drag calculations were of specific concern because the connecting steel wire only has a diameter of 2 mm. As an illustration Figure 4a shows the resulting total body pressure calculated by the CFD simulation for flight conditions in the upper troposphere (i.e. here 25 000 ft) at aircraft speeds near 172 kt (319 km h$^{-1}$). The highest total pressure regions occur in the front of the CCP instrument and at the edges of the tail stabilizers in the rear part of AIRTOSS. Regimes with a lower total pressure indicate flow conditions associated with lower turbulence level associated with the flow acceleration. Figure 4b provides an example of the typical velocity distribution around the AIRTOSS body. The deceleration zone as identified by Weigel et al. (2016) in the region of the CCP measurement volume corresponding to its front surface can be well discerned on the left side of the graph. The acceleration regions (red colored areas) originating from the streamline curvature effects follow. Figure 4c shows an iso-surface of the turbulent kinetic energy with a value of $150\,\text{m}^2/\text{s}^2$ colored by the velocity magnitude. Here the highest speeds occur downstream of the CCP's measurement volume. As an overall result of the CFD simulations the horizontal tail stabilizers of the AIRTOSS body were modified by affixing small air brakes to them



in suitable positions such that the asymmetry effects of the CDP were fully compensated (see Figure
3b). Accordingly during level flights the AIRTOSS moved quietly in the flow, without disturbing
oscillations, and the stable attitude necessary for the radiation measurements was maintained well.

### 2.6   Trace gas instruments

Besides the radiation and microphysical instruments, the AIRTOSS-Learjet tandem platform was
equipped with a suite of instruments quantifying the concentration of different trace gases.

The Fast Aircraft-Borne Licor Experiment (FABLE) was integrated on the Learjet to detect the
amount of carbon dioxide ($CO_2$) at flight altitude (Gurk et al., 2000). Nitrous oxide ($N_2O$) and
carbon monoxide (CO) were measured with the University of Mainz Airborne QCL-Spectrometer
(UMAQS, see Mueller et al. (2015) for more details).

Temperature and relative humidity measurements were made on the Learjet and on the AIRTOSS

by the MOZAIC Capacitative Hygrometer (MCH) which belongs to the Measurement of OZone by
AIRBUS In-Service AirCrafts (MOZAIC) system. The MCH uses a capacitative sensor and a Pt100
element to measure the relative humidity and the temperature respectively. The accuracy is $\pm 0.5\,°C$
for the temperature measurement and $\pm 5\,\%$ for the detection of the relative humidity. Evaluation-
and measurement-methods of the MCH are described in detail in Neis et al. (2015).

Water vapor measurements were taken by the Fast In-Situ Hygrometer instrument (FISH) and the
Selective Extractive Airborne Laser Diode Hygrometer (SEALDH). The FISH instrument is devel-
oped and operated by the *Forschungszentrum Jülich*. It is based on Lyman-Alpha-Photometry and
detects water vapor in a range between 1 ppmv and 1000 ppmv with an uncertainty of $\pm\,0.2$ ppmv
(Zöger et al., 1999). SEALDH is operated by the *Physikalisch-Technischen Bundesanstalt* and uses

Tunable Diode Laser Absorption (TDLAS) to estimate the concentration of water vapor in the at-
mosphere. It operates in a detection range between 25 ppmv and 25 000 ppmv with an uncertainty of
$<2\,\%$ and a time resolution of $<1$ s (Buchholz et al., 2013).

Ozone ($O_3$) measurements were performed on the Learjet by using a UV-Photometry 42 M Ozone
Analyzer developed by *Environment S.A.* This instrument detects the UV-absorption caused by $O_3$ at

a wavelength of 254 nm in a measurement range between 0 ppb and 10 000 ppb with an uncertainty
of 10 % (Köllner, 2013).

### 3   Results from the cirrus measurements during AIRTOSS-ICE

On 4 September 2013, the northern part of Germany was located between a high pressure system
with its center above southern Germany and a low pressure system above Scandinavia. A related

warm front in combination with cirrus passed the measurement area above the Baltic Sea. The cirrus
deck was probed by the AIRTOSS-Learjet tandem platform between 09:10 UTC and 09:40 UTC.
The observations indicated that the cirrus was located at an altitude between 8100 m and 10 200 m





with temperatures between -30 °C and -46 °C. Ice particle number concentrations of up to $1.4\,\mathrm{cm^{-3}}$ were found in several patches by the CCP in the upper cloud layer ($> 9000\,\mathrm{m}$) where temperatures

ranged below -40 °C. As discussed by Kärcher and Lohmann (2002), these high ice particle number concentrations only occur with vertical velocities higher than $30\,\mathrm{cm\,s^{-1}}$. Updrafts in warm fronts typically have vertical speeds less than $10\,\mathrm{cm\,s^{-1}}$ (Heymsfield, 1977) and cannot explain these high ice particle number concentrations. It appears that local convective cells with stronger updrafts lifted droplets from lower cloud layers to the cirrus altitude. As a result, homogeneous freezing in the cir-

rus environment might have been initiated and explain the high ice particle number concentrations in the upper part of the cirrus.

### 3.1    Microphysical measurements

The flight paths of the AIRTOSS and the Learjet are shown in Figure 5. The color coded line in Fig-

ure 5a shows the mean ice particle diameter measured by the CCP-CIPg. For each altitude a mean particle number size distribution was calculated. The flight sections at constant altitude that were used for the averaging are marked in Figure 5a. The legs were executed on constant altitude levels and are longer in the lower part of the cloud to get appropriate counting statistics for the optical particle instruments. Figure 6 displays the corresponding particle number size distributions and 2D

shadow images, detected by the CCP, for every single flight leg. The total particle number concentration $N$ is provided in the left panels and shows a typical increase with altitude from $0.26\cdot 10^{-2}\,\mathrm{cm^{-3}}$ ($8716\,\mathrm{m}$) to $8.4\cdot 10^{-2}\,\mathrm{cm^{-3}}$ ($9939\,\mathrm{m}$). Also, the particle size corresponding to the maximum of the size distributions shifts with increasing altitude from $300\,\mathrm{\mu m}$ ($8716\,\mathrm{m}$) to $30\,\mathrm{\mu m}$ ($9939\,\mathrm{m}$). The decrease in particle diameter with increasing altitude is also obvious in the 2D shadow images (right

panels in Figure 6). Higher ice particle number concentrations with small particle diameters in the upper cloud layers and lower ice particle number concentrations with large particle diameters in the lower cloud layers are typical for frontal, midlatitude cirrus and result from the microphysical process during the formation of the cirrus. As long as the relative humidity with respect to ice is sufficiently high, the particles start to grow by diffusion, gain mass and descent. This sedimentation

process leads to a redistribution of the ice particles inside the cirrus, with higher particle concentrations at cloud top. Nevertheless, the irregular particle shapes of the 2D shadow images in the lower part of the cirrus indicate that aggregation could also be a possible particle growth process. To analyze if diffusion or aggregation is the dominant process inside the observed cirrus, similar to Heymsfield and Westbrook (2010), terminal velocities were calculated. As an example, a spherical

(area ratio = 1) and a column shaped (area ratio = 0.25) ice particle with an initialized diameter of $D_p = 200\,\mathrm{\mu m}$ are assumed. This represents the measured conditions during Flight Leg 3 at an altitude of $9333\,\mathrm{m}$ (see Figure 6). For the spherical particle, a terminal velocity of $v_t = 91\,\mathrm{cm\,s^{-1}}$ was calculated, while for columnar particles $v_t = 14.5\,\mathrm{cm\,s^{-1}}$ was estimated. With these numbers, the



particles would need 11 minutes and 71 minutes, respectively, until they reach the bottom layer of
the cloud at an altitude of 8716 m. Following the discussion by Kienast-Sjögren et al. (2013), parti-
cles with a number concentration of $5.8 \cdot 10^{-2}\,\text{cm}^{-3}$ (Level 3 in Figure 6) need at least several hours
before aggregation processes occur. For this reason, aggregation is unlikely, and diffusional growth
seems to be the dominant process for this particular cirrus observed during AIRTOSS-ICE.

## 3.2 Solar downward irradiance

In addition to the microphysical measurements, collocated measurements of spectral solar radiation
were performed during the cirrus event of Section 3.1. Similar to Figure 5a, a profile of the spectral
downward irradiance (at 670 nm wavelength) measured by SMART on AIRTOSS and Learjet is
given in Figure 5b. The individual legs were filtered for turns of both platforms which assures that
only level flight conditions were considered. Additionally, only legs flown in the same direction
and above the same locations were chosen to assure similar cloud and surface conditions below the
cirrus. In total, five legs with simultaneous measurements on AIRTOSS and the Learjet are available
with larger vertical separation in the cirrus and less separation at cloud top and above. The impact of
the cirrus on the downward irradiance is most obvious in the two lower legs where the radiation is
attenuated by the cirrus. The attenuation is highly variable due to the horizontal heterogeneity of the
cirrus. However, both sensors on AIRTOSS and Learjet show almost the same pattern, illustrating
the collocation of the measurements. The similarity in the two datasets also results from the small
vertical displacement of Learjet and AIRTOSS of less than 200 m. During the higher flight legs,
the attenuation of downward irradiance by the cirrus is significantly lower. In the third leg, only
AIRTOSS measurements are slightly affected by the cirrus, while the Learjet already observed clear
sky conditions. Above the cirrus, the downward irradiance is almost undisturbed and constant over
the entire legs.

## 4 Analysis

Two cases are selected to illustrate the potential of the collocation of measurements achieved by
the AIRTOSS-Learjet tandem platform. Due to the different instruments operated on AIRTOSS and
Learjet different combined analysis of data are possible. Beside combining in-situ and radiation
measurements also the simultaneous radiation measurements on both platforms can be analyzed
jointly.

### 4.1 Collocation of microphysical and radiative properties

Figure 7 shows a time series of downward spectral irradiance at 670 nm wavelength measured from
the Learjet (Figure 7a) and AIRTOSS (Figure 7b) along a flight leg observed on 04.09.2013 between





09:35 UTC and 09:39 UTC. In addition, Figure 7c shows the detected number concentration of the CCP-CDP and the CCP-CIPg. This data was obtained from a flight leg, when the AIRTOSS operated at an altitude of around 9900 m. The cloud particle number concentrations above zero were detected

within two sections of the flight leg and indicates that AIRTOSS did penetrate two cirrus filaments at the top of the cirrus layer. The downward irradiance has been constant for most of the flight leg indicating clear sky conditions without attenuation of the incoming solar radiation. The strongest deviation from the clear sky conditions was found at about 09:38:05 UTC where the irradiance shows a rapid decrease for both platforms. This coincides with higher values in the particle number con-

centration measurements. The increasing number concentration indicates that AIRTOSS is located in a thicker part of the sampled cloud and certainly the cloud top is above AIRTOSS. As the Learjet measurements are located closer to cloud top the effect is here smaller compared to the AIRTOSS observations. At cloud edges also an increase of the irradiance can occur due to three-dimensional radiative effects. For the smaller cloud observed at the beginning of the leg (09:35:45 - 09:36:40 UTC),

only the downward irradiance measured by the AIRTOSS is affected, while the downward irradiance measured by the instruments on the Learjet remains almost constant. At this time only AIRTOSS was located inside the cirrus while the Learjet flew above cloud top and consequently only the downward radiation in the altitude of AIRTOSS was reduced.

Such constellations are well suited to investigate the interaction of cloud microphysical and radia-

tive properties as demonstrated by Werner et al. (2014) for shallow cumulus. However, this analysis of collocated number concentration and downward irradiance measurements works only for investigations of the uppermost cirrus layer. The cirrus investigated here showed a vertical extension of approximately 2100 m. As soon as both, the Learjet and the AIRTOSS, were completely inside the cirrus, the inhomogeneities above and below the tandem platform dominate the measurements and a

correlation between the microphysical and the radiative measurements is no longer evident.

### 4.2 Vertical profile of solar heating rates

The spectral irradiance measurements were integrated to broadband quantities and averaged for the individual horizontal legs as indicated in Figure 5. The change of the solar position in between measurements of the different legs was taken into account by normalizing the irradiance to observations

from the uppermost level. Figure 8a shows the corresponding vertical profiles of upward and downward broadband irradiance measured on AIRTOSS and Learjet. Horizontal bars indicate the standard deviation along an individual leg as well the variability of the radiation along the flight legs.

The upward irradiance varies significantly with altitude albeit without showing a regular pattern. This is caused by slight changes of the flight track and in the cloud situation; mainly the presence

of a low stratus cloud below the cirrus. The standard deviation of upward irradiance indicating the cloud variability is higher in the upper three legs, while the two lower legs show less variability when the sensors are located well below cloud top.





The profile of downward irradiance also indicates the presence of a cirrus. While above cloud top
the values remain vertically constant and show only a small standard deviation, larger variability and
a decrease of the downward irradiance is observed when the instruments enter the cloud. Upward
and downward irradiance $F^\downarrow$ and $F^\uparrow$ at two different altitudes, $z_1$ and $z_2$ are used to calculate the
effect of the radiation field on the local temperature change in terms of heating rates at a certain
altitude $z = 1/2 \cdot (z_1 + z_2)$. The heating rate $\partial T/\partial t|_z$ in units of $\mathrm{K\,day}^{-1}$ within the layer is derived
following Wendisch and Yang (2012)[Eq. 9.66]:

$$
\begin{aligned}
\left.\frac{\partial T}{\partial t}\right|_z &= \frac{1}{\varrho \cdot c_p} \frac{\partial F_{net}(z)}{\partial z} \\
&\approx \frac{1}{\varrho \cdot c_p} \cdot \left[ \frac{F_{net}(z_2) - F_{net}(z_1)}{z_2 - z_1} \right] \\
&\approx \frac{1}{\varrho \cdot c_p} \cdot \left\{ \frac{\left[ F^\downarrow(z_2) - F^\downarrow(z_1) \right] - \left[ F^\uparrow(z_2) - F^\uparrow(z_1) \right]}{z_2 - z_1} \right\} .
\end{aligned}
\tag{1}
$$

Figure 8b shows profiles of $\partial T/\partial t|_z$ derived in two different ways. Assuming only a single aircraft is
available, the solar heating rates can be calculated by the irradiance profile measured by this single
aircraft, either AIRTOSS (red circles) or Learjet alone (blue circles). Having the combined collo-
cated measurements of both, AIRTOSS and Learjet, heating rates can additionally be derived along
each horizontal leg (black circles). The heating rate profiles obtained for the investigated cirrus sig-
nificantly differ depending on the chosen method. To interpret these differences, uncertainties of the
heating rates were calculated for both approaches. An uncertainty of 6 % in the radiometric calibra-
tions was assumed which directly propagates into the calculated heating rates (Eq. 1) as all sensors
are calibrated identically. All remaining uncertainties of the irradiance are estimated with 0.5 %. For
the single aircraft approach the irradiances are always measured with the same system. This reduces
the impact of the remaining uncertainty to contributions of the two net irradiance only. In the collo-
cated approach, two independent systems are used and all four irradiance measurements contribute
to the overall uncertainty. Additionally, the distance $z_2 - z_1$ influences the accuracy of the heating
rate. Due to the geometry and the flight altitudes, this distance differs for both approaches. Larger
distances between the two measurements provide more precise results. While $z_2 - z_1$ amounts about
200 m for the collocated approach, determined by the length of the wire between AIRTOSS and the
Learjet, $z_2 - z_1$ of the single aircraft approach depends on the altitudes of the legs and is typically
larger (500 m at cloud bottom and 300 m at cloud top). Overall, the uncertainty of the heating rate
estimates derived from the collocated approach theoretically are expected to be significant larger
than for the single aircraft approach. However, although the profiles using only AIRTOSS and only
Learjet data agree with each other, the profiles show large scatter with heating rates ranging from
-13 $\mathrm{K\,day}^{-1}$ to +33 $\mathrm{K\,day}^{-1}$. These unrealistic heating rates mainly result from changes in the up-
ward radiance between two individual flight legs. As the legs are not perfectly collocated and a low



stratus layer did change its location below the cirrus during a flight level change (~2 min temporal separation), the data set is not consistent and leads to incorrect heating rate estimates.

By contrast, the collocated data set does not suffer from changing conditions below the cirrus as
both sensors always observe the same scene at the same time. Consequently, the heating rate profile in Figure 8b does show a smoother and more realistic pattern with values always ranging between $0\,\mathrm{K\,day^{-1}}$ and $6\,\mathrm{K\,day^{-1}}$, which are typical values for a thin cirrus.

This improvement in calculating heating rates illustrates the benefit of collocated irradiance measurements. However, the derived heating rates still do not represent theoretical results as provided
by e.g., Bucholtz et al. (2010) and Thorsen et al. (2013). This may be caused by horizontal inhomogeneities of the observed cirrus leading to horizontal photon transport as discussed by Finger et al. (2016).

## 5   Conclusions

The AIRTOSS-ICE campaign was conducted to perform collocated measurements of cirrus clouds by using the advanced AIRTOSS-Learjet tandem platform. A combination of the Learjet and the AIRTOSS, both equipped with the SMART sensor and in-situ instruments, allowed for measurements of radiation and microphysical properties on different altitudes using just one aircraft. The new certification for the AIRTOSS-Learjet tandem platform made it possible, for the first time, to
probe cirrus clouds at altitudes up to 36 000 ft with the new measurement package. The campaign successfully showed that collocated measurements with the further, developed AIRTOSS-Learjet tandem platform provide useful information. This is demonstrated by combining the microphysical and radiative measurements and by deriving solar heating rates.

Using AIRTOSS-ICE measurements vertical profiles have been derived, which showed that heating
rates can be estimated with higher accuracy when collocated measurements are applied instead of using a single platform. Despite the theoretically higher uncertainties introduced by the measurement errors from two independent measurement systems, the collocated observations resulted in a more realistic profile of heating rates as these are not affected by changes of the radiation field below the observations altitude (e.g., inhomogeneous surface albedo, lower cloud layers). Observations per-
formed with a single aircraft strongly depend on stable conditions between consecutive flight legs and, therefore, are subject to serious uncertainties in derived profiles of heating rates.

However, AIRTOSS-ICE also showed the limits of the collocated measurement setup. The investigated cirrus had a thickness of more than 200 m, which is larger than the distance between Learjet and AIRTOSS. This did not allow for the radiative instruments to measure concurrently with the
AIRTOSS below and with the Learjet above the cirrus layer, which would have been needed to derive the cirrus radiative layer properties (Finger et al., 2016). The short distance between both





platforms resulted in only small differences in the upward and downward irradiances measured on the AIRTOSS and the Learjet for this proof-of-concept campaign and was a compromise between the scientific interests and the manageability of the platforms. An increase of the vertical distance beyond 200 m is also not easy to achieve. It would require a longer steel wire and/or a slower aircraft, as well larger areas where such flights are permitted. For clouds with a larger vertical extent, two single aircraft could be a better choice. It certainly depends on the scientific goals and instrumentation whether or not the AIRTOSS-Learjet tandem platform is the appropriate choice.

With respect to microphysical inhomogeneities, the vertical separation of 200 m between both platforms is sufficient for cirrus studies. What would be required are microphysical in-situ instruments with overlapping measurement characteristics, or, ideally, two identical instrument sets on both platforms. To perform microphysical measurements with a higher temporal resolution, the implementation of holographic instruments is also an attractive alternative. These instruments have a larger sample area (3.6 x 2.4 cm$^2$) and a higher sampling rate (Schlenczek et al., 2016). Furthermore, the integration of trace gas instruments both inside AIRTOSS and the Learjet could be used, e.g., for collocated trace gas measurements in the vicinity of the tropopause layer, the edges of tropopause folds, streamers etc. To study different atmospheric conditions or to obtain better statistics of cirrus cloud, the operation of the AIRTOSS-Learjet tandem platform in other regions, outside of military restricted areas, will be a significant challenge. This could be accomplished in less populated regions, like the polar regions, remote areas of the oceans, rain forests and others.

*Acknowledgements.* The AIRTOSS-ICE project was supported by the *Deutsche Forschungsgemeinschaft* (DFG) through projects "WE 1900/19-1, BO 1829/7-1, SP 1163/3-1" and on a significant level by internal funds of the *Particle Chemistry Department at the Max Planck Institute for Chemistry*. We particularly thank the pilots and the crew of the *Gesellschaft für Flugzieldarstellung* for making this project possible. We are also thankful for the support of the electrical engineers Wilhelm Schneider and Christian von Glahn (University of Mainz) and all other participants of the AIRTOSS-ICE campaign.



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



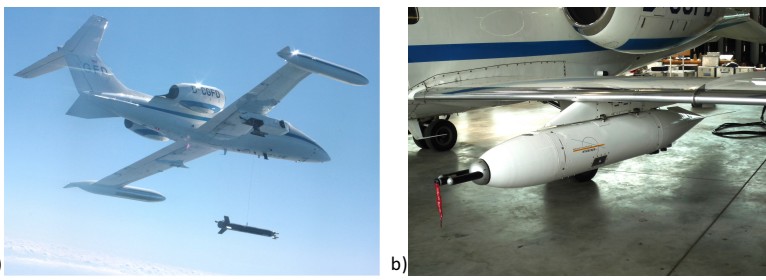

**Figure 1.** (a) advanced AIRTOSS-Learjet tandem platform: Learjet 35A with the sensor shuttle (called AIR-TOSS) during a test flight. The photograph was taken during the release of AIRTOSS. When AIRTOSS is fully released, the distance between Learjet and AIRTOSS is 3000 ft (914 m). (b) attached sensor pod under the left wing of the Learjet with the mounted FSSP at the tip.

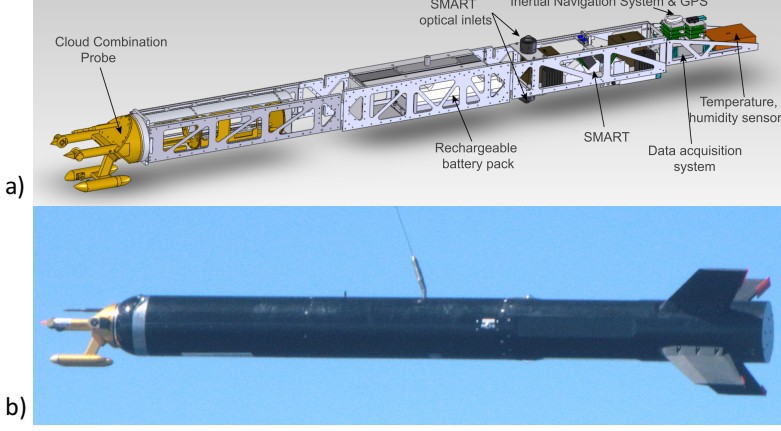

**Figure 2.** Different states of the AIRTOSS development process. (a) shows a perspective view with the position of the instruments (Röschenthaler, 2013), including the Spectral Modular Airborne Radiation measurement sysTem (SMART). (b) shows the manufactured AIRTOSS during a mission.





| | Component | Mass in kg | Explanation |
|---|---|---|---|
| **Front** | CCP | 9.10 | Cloud Combination Probe (2 - 960 µm particle diameter) |
| **Middle** | Rechargeable battery | 10.8 | Power source for all instruments |
| **Rear** | Radiation optical inlet | 0.24 | 4 pieces, top and bottom |
| | Spectrometer (near infrared) | 0.56 | 2 pieces, near infrared spectrometer (1000 - 2200 nm, 9 - 16 nm resolution) |
| | Spectrometer (visible) | 1.75 | 2 pieces, visible spectrometer (300 - 1000 nm, 3 nm resolution) |
| | Peltier-Element | 0.33 | 2 pieces |
| | INS | 0.02 | Inertial Navigation System |
| | GPS-Sensor | 0.04 | Global Positioning System |
| | Rosemount + Sensors | 0.60 | Temperature and humidity measurements |
| | ICH-TB | 0.40 | Temperature and humidity measurement electronics |
| | Power supply BEP-5150C | 0.75 | Power supply (12V, 5V) |
| | Computer | 1.26 | Data acquisition |
| | Shutter | 0.10 | 2 pieces, for SMART-System |
| | Shutter-Control | 0.13 | 2 pieces, to control the shutters |

**Table 1.** Masses of the different instruments and their accessories, mounted inside AIRTOSS.

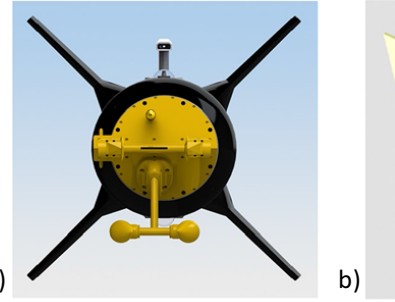
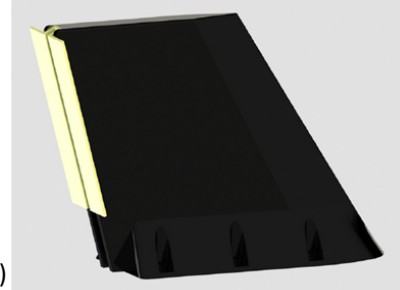

**Figure 3.** (a) front view of the AIRTOSS showing the asymmetry shape of the CCP instrument. (b) air brake at one wing of the AIRTOSS (Röschenthaler, 2013).





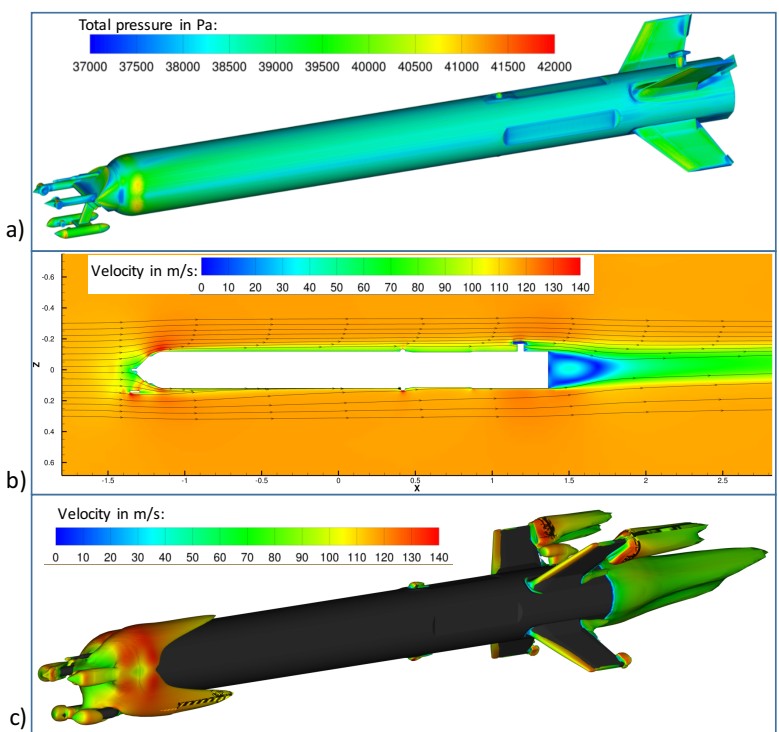

**Figure 4.** Flow simulations for flight conditions: (a) resulting total body pressure, (b) velocity distribution around the AIRTOSS body, (c) shows an iso-surface of the turbulent kinetic energy with a value of $150 \, \mathrm{m^2 \, s^{-2}}$ colored by the velocity magnitude (Röschenthaler, 2013).

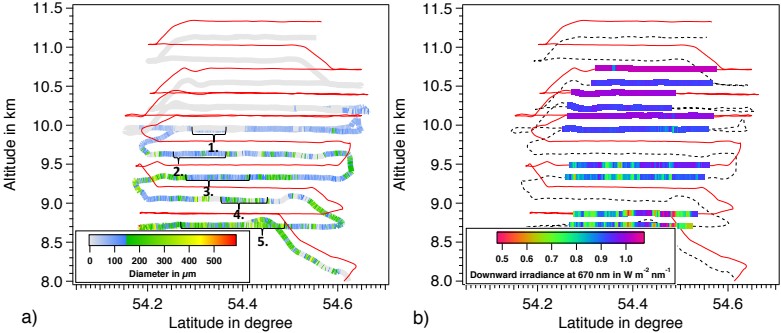

**Figure 5.** Both panels show the flightpath of the Learjet (red line) and the flightpath of AIRTOSS (dashed line) overlain by color coded measurements of particle mean diameter (panel a) and downward irradiance at 670 nm (panel b). The flight sections used to calculate the leg mean particle diameter are indicated in panel a.





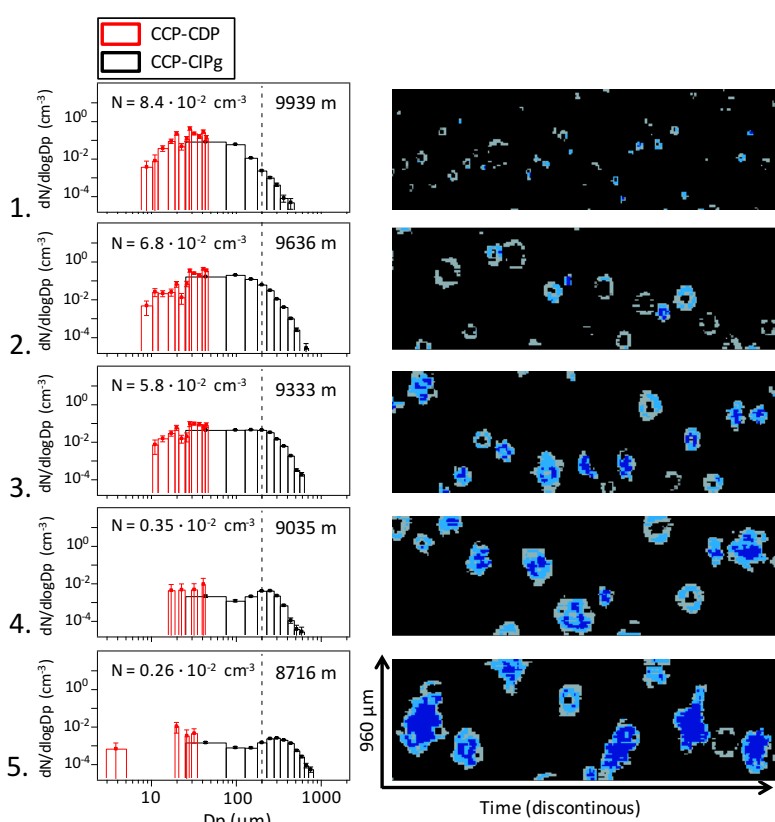

**Figure 6.** Microphysical characteristics of the marked flight legs from Figure 5a. Left panel: Combined size distributions of the CCP-CDP (red) and the CCP-CIPg (black) instrument mounted on the AIRTOSS. With an increasing altitude, the maximum of the size distribution shifts to smaller particle diameters. Right panel: Sample 2D shadow images from every single flight leg, recorded by the CCP-CIPg instrument. The different colors represent the shadow intensity (grey > 35 %, light blue > 50 %, dark blue > 65 %).





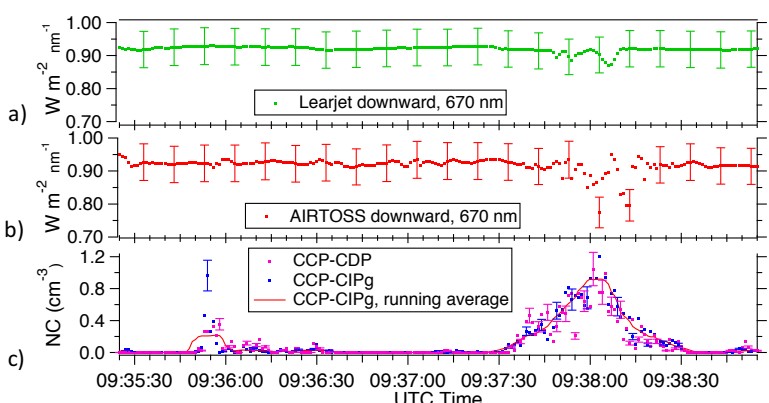

**Figure 7.** Downward spectral irradiance at 670 nm measured from the Learjet (a) and the AIRTOSS (b) and number concentration (NC) measured on the AIRTOSS platform with the CCP-CDP (2 – 50 µm) and the CCP-CIPg (15 – 960 µm) instrument (c). The data was obtained at the highest flight leg, measured on 04.09.2013, where the AIRTOSS flow at an altitude of around 9900 m.





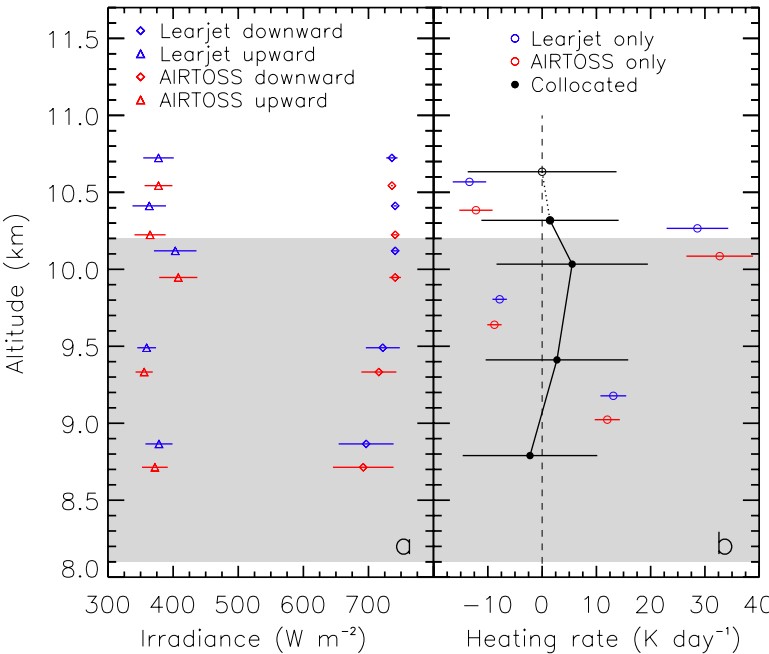

**Figure 8.** a) Profiles of vertical upward and downward broadband irradiance measured on AIRTOSS and the Learjet. The bars indicate the standard deviation of the irradiance along the individual flight legs. b) Solar heating rates calculated from the irradiance profile either using a single platform or the collocated measurements. The gray area indicates the cirrus layer as indicated by the CCP.