# Peer review of "A tandem approach for collocated in-situ measurements of microphysical and radiative cirrus properties"

_Atmospheric Measurement Techniques, 2017_

## Referee Comment (RC1)

**Full review of Klingebiel et al., AMT 2017 (based on manuscript version of 14 Feb 2017)**

**General comments**

The study titled "A tandem approach for collocated in-situ measurements of microphysical and radiative cirrus properties" by M. Klingebiel et al. describes how a tandem measurement platform consisting of a research aircraft and a retractable towing sensor shuttle equipped with a CCP and solar radiation instruments can be used to obtain vertical profiles of microphysical and radiative cirrus properties. Spatial cirrus inhomogeneities of properties such as particle size corresponding to the maximum of the particle size distributions were quantified for one case study. Also, it was shown that solar heating rates derived from collocated measurements of the tandem platform lead to more realistic values than those based on single instruments. Finally, the limitations of the tandem platform were listed.

While the approach is unique and worth publishing the quality of the writing needs improvement. Besides assuring a proper English grammatical sentence structure, the task of the main author should be to harmonize different pieces of information to make a coherent story. Especially the abstract would benefit from presenting the information more concisely and in a more logical order. Also, a wrap-up sentence in the abstract summing up the results or their implications is missing. Additionally, you mention ten flights in the abstract which leads to anticipation of results of ten flights which are not fulfilled.

I would suggest the manuscript to be published after minor revision. The authors should address the following points:

**Major comments**

Line 110-112: You mention that on the original AIRTOSS, the external body cover was used as a mounting point for additional payload. Please explain why this was modified.

Line 114: Air brakes are the red rectangles on the winglets in the back? This becomes clear only later on. – Describe the photo more clearly to a reader who might not know what air brakes are. Also, did you have several different flights during which you employed air brakes with different resistance coefficients to see which lead to the best performance in terms of horizontal flight positioning? Or did you construct the air brakes after flow simulations? ...ok, some of this is answered in Section 2.5 – you can also mention in line 114 that details are explained later. But if you don't, the reader is lost.

Line 137: You mention that several heaters of the CCP were deactivated. – Mention if/how this measure affects the instrument performance?

Line 335-353: This paragraph should be structured and phrased more clearly. For readability, it is better to introduce it like For flight X from Y to Y UTC, with the aircraft flying at XX m altitude and the AIRTOSS being at YYm altitude, cirrus filaments were detected during two sections (at X UTC and Y UTC). ...then go into detail. Instead of starting with details and then giving the big picture in the end. Also, in Fig.7a,b the quantity measured (downward irradiance needs to be added in the y-label). Axis labels and legend font is too small. Do the vertical bars indicate errors or standard deviations? What is the temporal resolution of the measurements?
In Fig. 7c an increased NC (of CCP-CDP and CCP-CIPg) is obvious at 05:35:50UTC – why does the running average only increase a few seconds later. – How is the running average determined?

Line 368-371: In this paragraph you mention that variation in the upward irradiance is mainly due to a lower level stratus cloud. You also state that the upward irradiance varies more strongly in the upper legs while it is less in the lower legs. – Shouldn't the influence of the underlying stratus be affecting the lower leg measurements more than the upper ones? – Please clarify. Also, an additional figure showing a satellite image with overlaid flight track would be good to illustrate the cirrus/stratus situation.

Lines 405-410: This is important! – It should be mentioned more clearly in the abstract. Please emphasize that only collocated irradiance measurements of the Learjet and the AIRTOSS give meaningful heating rates. Also, specify which heating rates are theoretically expected instead of only listing the corresponding references.

Line 407: Here you mention that a cirrus geometrical thickness of more than 200m is too large to allow for positioning of the Learjet above and the AIRTOSS below the cloud layer. Earlier you stated a longer steel wire length – please clarify why the AIRTOSS cannot be positioned below thicker clouds?

Line 427-428: What exactly can you derive by combining microphysical and radiative measurements. You did show several graphs of collocated measurements but it become not quite clear how this knowledge can be used. – Is it possible to validate radiative transfer retrievals of particle size (based on measured radiative properties) with the simultaneously measured particle size distributions? Or how else can the measurements be used for more in-depth cirrus studies?

Line 443-448: Only here you mention that the shown results are taken from a proof-of-concept campaign and that thus the AIRTOSS steel-wire was not extend further. – Please mention that in the very beginning of the manuscript.

Section 2.6: The trace gas measurements seem totally unrelated to the paper in which you are focusing on collocated measurements microphysical and radiative

properties. Unless you convince me how they add to the entire story, I would suggest to remove the parts referring to the trace gas measurements. You only briefly refer to the trace gas measurements again in lines 455-457. – This is not sufficient to justify the inclusion of the trace gas measurement description.

**Minor comments**

Sometimes you refer to the towing sensor shuttle as AIRTOSS, sometimes as the AIRTOSS. Be consistent and choose if you want to call it a noun or if you want to refer to it as proper name.

Line 4: "detached from" should be extended by "detached from the aircraft via a cable" to illustrate the setup more clearly
Line 6: replace "layer clouds" by the more scientific term "stratiform clouds"
Line 6: motivate why you need "sophisticated numerical flow simulations"  - to quantify shattering effects on the CCP?
Line 9-10: move this sentence about the steel cable to line 4 for clarity
Line 13 (and 287): The sentence seems backwards: ice crystals grow from small to large sizes (via diffusional growth/aggregation), thus the sentence should be phrased: …maximum size in the observed…increases from 30mum to 300mum with decreasing altitude. Also, shouldn't the change in maximum size of the PNSD rather refer to geometrical cloud depth than merely altitude? Please clarify.
Line 16: Remove "consequently" or replace it by "thus"
Line 16:  Add "growth" between microphysical  and process
Line 17: is the solar downward irradiance on the Learjet measured above/in/below the cirrus?
Line 18: Clarify where the cloud is positioned with respect to the tandem platform to determine heating rates
Line 25: THEIR microphys. Prop. ; warm or cool (plural!)
Line 26-28: rearrange sentence structure to proper English. "Especially the ice particle shape was found to determine … (e.g., Wendisch … )"
Line 29: You cannot talk about "such effects" of surface roughness when you haven't previously talked about surface-roughness. – Modify the sentence accordingly.
Line 47: Clarify if the "two helicopter borne platforms" refer to two helicopters flown simultaneously or if not, what kind of platforms you refer to.
Line 54: Replace "speed" by "aircraft velocity"
Line 55: released by means of a steel wire
Line 56: In "the study of" Frey et al….
Line 58: "this" not "his"
Line 60: If the Frey et al. 2009 study is based on the proof-of-concept campaign, it should be mentioned clearly. Also, the proof-of-concept sentence should be moved before line 56. Try to ease the reader into the subject, go from larger picture to more detailed description.
Line 94: What is the limited distance? Give a value.
Line 103: Title of this subsection should be "Specifications of the AIRTOSS"
Line 113: remove comma

Line 121: "of up to 914m"
Line 128: "less than the maximum …"
Line 137: to save energy
Line 138: explain abbreviation CCP-CDP
Line 139: a voltage
Line 141: no commas
Line 153: mounted on
Line 154: Seems like a word is missing after particle-by-particle data → analysis/algorithm/technique?
Line 158: Specify what you mean by size: maximum dimension?
Line 163: citations should be given in chronological order
Line 172: Again, this last sentence seems like it was added as an afterthought. Consider moving it after the reference to Knollenberg, maybe by combining those two sentences.
Line 178: at the bottom
Line 180: wavelengths
Line 180: irradiance sensor; give reference for horizontal alignment requirement
Line 191: …symmetric, … (comma)
Line 194-197: this sentence needs to be simplified or devided into two for clarity. What do you mean by "aiming at their compensation"?
Line 219: As a result, …
Line 235: Accordingly, …
Line 272: of less than…
Line 293: growth process
Line 294: water vapor diffusion; the particles don't descent, they sediment
Line 300: explain the term area ratio
Line 304: what orientation was assumed for the falling columnar ice crystal?
Line 304: replace numbers with "estimated terminal fall velocities"
Line 307: Why does aggregation only occur several hours after particle formation at such ice particle number concentration? – Try to present the reader with a good story, instead of with many questions.
Line 326: What do you mean by "undisturbed"? constant?
Line 349: add citation
Line 350: is affected by what? Do you mean "shows variation"?
Line 359: the "in-cloud" inhomogeneities
Line 363: Start the sentence with "to make measurements comparable, …"
Line 367: Sentence is unclear. Please clarify what the horizontal bars indicate: the standard deviation along individual flight legs or the variability of the radiation along the flight legs?
Line 406: why radiance? I suppose you mean "irradiance"?
Line 420: Is SMART really a sensor?
Line 426: Remove comma
Line 454: Again, the reader wonders: What is the higher sampling rate? – Please mention it and relate to the sampling rate and the sample area of the CCP.

---

## Short Comment (SC1) · 26 May 2017

Dear Authors ..

first of all congratulations to very interesting experimental approach and to a great manuscript

With regard to the description of the use of the SEALDH instrument during the AIRTOSS campaign (line 250 to 257 of the manuscript) i have however a few comments and background informations which i think should be (at least partially) incorporated into the manuscript :

The SEALDH instrument development was started in Heidelberg University, Germany

and went on at Physikalisch-Technische Bundesanstalt , PTB. In the course of this development several versions of SEALDH were developed (Main versions 0, I and II), which actually have different performance parameters with respect to accuracy, sensitivity and concentration range. It is therefore necessary in the manuscript to clearly state which instrument version was used, to update the relevant performance parameters in the article, and to cite the suitable articles as well as the reference for the most recent SEALDH version.

The instrument which was flown during AIRTOSS shouldn't be named "SEALDH". The correct version of SEALDH flown during AIRTOSS was (a first version of) "SEALDH-II". This should be adopted in the manuscript. Further we would recommend to add/replace the description of SEALDH and incorparate an extract of the relevant information related to SEALDH-II.

SEALDH-II's measurement uncertainty is calculated based on a physical model of the instrument (Buchholz et al., 2016 > SENSORS). The measurement technique used is correctly named as dTDLAS (direct TDLAS) which indicates that the raw data evaluation is based on physical, first-principle model and thus leads without any previous gas-based instrument calibration to an absolute [H2O] concentration value. In this proprietary evaluation mode SEALDH-II has a 4.3% linear and a ± 3 ppmv offset uncertainty. Assuming an 11% maximum acceptable uncertainty, the design concentration range of SEALDH-II starts at about 30 ppmv and reaches to near the water vapor condensation point in the instrument, which in the current version is reached when the dew point temperature gets close to the instrument temperature (this leads roughly to 40 000 ppmv). The long-term stability of the most up to date version of SELADH II was recently validated over a consecutive period of 18 month with respect to the highest metrological humidity standard of Germany at PTB. This validation indicated (relative to the primary standard) an average offset term of 0.17% and an average scatter of 0.35% (1). In terms of the procedures described in the paper by Klingebiel et al , this number indicate the "accuracy/uncertainty" of SEALDH-II, und the assumption

SEALDH had been calibrated using the primary standard data. SEALDH-II resembles a new hygrometer generation, as it is the first calibration-free airborne hygrometer with a direct primary metrological long term validation (Buchholz and Ebert, 2017» AMT . Hence AIRTOSS is the first airborne hygrometer campaign with a direct metrological linkage to a primary humidity standard.

The rest of this paragraph concerning SEALDH is fine (time resolution etc.), I would however recommend using the word "measure" rather than "estimate" the concentration.

Up to date references for SEALDH II: Buchholz, B. and Ebert, V.: SEALDH-II – a calibration-free transfer standard for airborne water vapor measurements: Pressure dependent absolute validation from 5 − 1200 ppmv at a metrological humidity generator, Atmospheric Measurement Techniques Discussions, (February), 1–22, http://dx.doi.org/10.5194/amt-2016-413, 2017.

Buchholz, B., Kallweit, S. and Ebert, V.: SEALDH-II—An Autonomous, Holistically Controlled, First Principles TDLAS Hygrometer for Field and Airborne Applications: Design–Setup–Accuracy/Stability Stress Test, Sensors, 17(1), 68, http://dx.doi.org/10.3390/s17010068, 2016.

---

## Referee Comment (RC2) · Anonymous Referee #1 · 9 Jun 2017

The paper demonstrates a new facility to measure vertically displaced but horizontally and temporally matched observations of radiation and microphysics.

The paper was well written. I could understand it all. Another proof read would capture the last of any outstanding typos and gramatical errors. I think that the paper would benefit from ending with a list of pros and cons about the system in terms of its scientific capability (e.g. improved heating rates) and operational deployment (e.g. only allowed in military flight areas).

In general I think the paper needs to emphasise the usefulness of the system more. I have one main point that it would be good to see resolved (see line 358-360 comment),

but otherwise i think that the paper is publishable subject to minor changes.

Specific points:

On reading the abstract i was not convinced why i needed to use this system. I think the paper needs to do a bit more to convince the reader that this is a useful technique.

line115. Does this mean that certification is limited to one payload and any changes require another certification?

section 2.6. It would seem more natural to move this section to just after section 2.3 or 2.4. Or move the flow simulation earlier. At the moment the flow simulation section sits in the middle of sections describing instrumentation.

line 260 0ppb - is it really that sensitive?

line 295 - do you mean smaller ice crystals nearer the top (lower fallspeeds and hence longer residence times at that altitude) ?

line 307. What was the relative humidity with respect to ice? Can you reconcile the 2D imagery in figure 6 with the diffusion grown images in Bailey and Hallett 2009 JAS fig5 for your temperature and humidity range?

line 358-360. This is the heart of the reason for flying a tandem formation. If you have one platform within cloud measuring the downwelling radiation and another platform slightly below measuring the same radiation then the difference between those two signals is going to provide information about the intervening cloud. It should not matter that one platform is not at cloud top. Perhaps the errors in the radiation measurements are too large to do this with the separation that was being used? Could a calculation be done to estimate what thickness is required?

It should now be possible to do a closure study where the microphysical information from AIRTOSS is assumed to represent a column of cloud between AIRTOSS and the Lear. An average along the leg could be used. This column can then be modelled with

a radiation code to estimate the effect on the radiation. The radiative response of this column of cloud can then be compared with the measured radiative difference. To me this would be the unique selling point of this system- the ability to carry out this type of analysis. This sort of closure study could be used to try and constrain unobserved quantities such as crystal roughness.

Fig8. Yes, this plot is good. The advantage of using the tandem platform for heating rates over single platforms should be emphasised more in the abstract.

---

## Author Response (AR1)

Authors response concerning the Manuscript prepared for Atmos. Meas. Tech.

**A tandem approach for collocated in-situ measurements of microphysical and radiative cirrus properties**

The Authors would like to thank the anonymous reviewers for their time as well as their very good suggestions and remarks. We think it improves this publication a lot.
We would also like to thank Volker Ebert for his comment about the instrument, which we included in the trace gas instrument section.

In the following, we answer all comments. Additionally, a marked-up manuscript version is attached.

*Questions* **and Answers regarding RC1:**

***Line 110-112: You mention that on the original AIRTOSS, the external body cover was used as a mounting point for additional payload. Please explain why this was modified.***

That is correct. We wanted to use the external body just as a cover because it made it easier to open the AIRTOSS to check the instruments and to exchange the battery. Besides this fact, it was much easier to arrange the instruments on an internal frame during the construction process.
We made it more clear in the paper by writing:

> *For the modified version, the body cover is used only as a cover, which does not need a detailed strength calculation and certification. It also makes it more convenient to access the instruments and to recharge the replaceable battery after a measurement flight.*

***Line 114: Air brakes are the red rectangles on the winglets in the back? This becomes clear only later on. – Describe the photo more clearly to a reader who might not know what air brakes are. Also, did you have several different flights during which you employed air brakes with different resistance coefficients to see which lead to the best performance in terms of horizontal flight positioning? Or did you construct the air brakes after flow simulations? ...ok, some of this is answered in Section 2.5 – you can also mention in line 114 that details are explained later. But if you don't, the reader is lost.***

Thank you for this comment. Yes, the air brakes are the red rectangles on the winglets in the back. They were constructed after the flow simulations, and we used one test flight to check

the behavior of the whole AIRTOSS. It turned out that the simulations were correct, and the AIRTOSS stayed incredibly stable during the flights.

We explained it more accurately in the text and refer to Section 2.5.

> *Air brakes (red rectangles at the winglets) with different resistance coefficients were mounted onto the winglets to compensate for the shape of the asymmetric CCP and to keep the released AIRTOSS in a horizontal flight position. More details about the air brakes and the associated flow simulations are given in Section 2.6.*

**Line 137: You mention that several heaters of the CCP were deactivated. – Mention if/how this measure affects the instrument performance?**

You are completely right, that was a big issue before the campaign. We needed to save as much power as possible to get at least an operating time of around two hours for the AIRTOSS. The heaters, which were deactivated, are usually important for avoiding icing at the tips of the CCP by flying e.g. through mixed phase clouds. Another reason for the heaters is to avoid condensation on the optics of the CCP. We expected that the air masses in the vicinity of cirrus are so dry that icing or condensation wouldn't occur. In Figure 6, it is visible that the electronics/measurements were not affected by icing or condensation, because plausible 2D shadow images measured by the CCP-CIPg are shown.

We added this information to the paper.

> *To save power, several heaters of the CCP instrument were deactivated. This was possible, because the main purpose of the heaters is to avoid icing and condensation at the optics of the instrument, by flying through e.g. mixed phase clouds. Only those from the CCP - Cloud Droplet Probe (CCP-CDP) instrument (see Section 2.3) were running during the measurement flights to keep the electronics under stable temperature conditions.*

**Line 335-353: This paragraph should be structured and phrased more clearly. For readability, it is better to introduce it like For flight X from Y to Y UTC, with the aircraft flying at XX m altitude and the AIRTOSS being at YYm altitude, cirrus filaments were detected during two sections (at X UTC and Y UTC). ...then go into detail. Instead of starting with details and then giving the big picture in the end. Also, in Fig.7a,b the quantity measured (downward irradiance needs to be added in the y-label). Axis labels and legend font is too small. Do the vertical bars indicate errors or standard deviations? What is the temporal resolution of the measurements?**

Thank you for this comment. We introduced the flight and the associated atmospheric conditions already in Section 3. For this reason, we didn't want to repeat it. Nevertheless, we agree with your remark and changed the first sentence, which now includes the date, the

time period of the flight leg and the altitude.

> *Figure 8 shows a time series of downward spectral irradiance at 670 nm wavelength measured from the Learjet (Figure 8a) and AIRTOSS (Figure 8b) during a flight leg observed on 4 September 2013 between 09:35 UTC and 09:39 UTC, when the AIRTOSS was operated at an altitude of around 9900 m.*

We changed the legend in Figure 7 to make it obvious that downward irradiance measurements are shown. Axis labels and legend fonts are bigger too. The vertical bars indicate the error of the instruments and the running average uses the boxcar smoothing algorithm with 10 repetitions. We added this in the description of the figure. The temporal resolution is 1Hz for all measurements.

***In Fig. 7c an increased NC (of CCP-CDP and CCP-CIPg) is obvious at 05:35:50UTC – why does the running average only increase a few seconds later. – How is the running average determined?***

As already mentioned in the previous answer, we used the boxcar smoothing algorithm with 10 repetitions. This explains the behavior of the smoothing, because the running average increases a few seconds earlier as the peak.

***Line 368-371: In this paragraph you mention that variation in the upward irradiance is mainly due to a lower level stratus cloud. You also state that the upward irradiance varies more strongly in the upper legs while it is less in the lower legs. – Shouldn't the influence of the underlying stratus be affecting the lower leg measurements more than the upper ones? – Please clarify. Also, an additional figure showing a satellite image with overlaid flight track would be good to illustrate the cirrus/stratus situation.***

That's right, our wording is a little contradictory and the explanation is not complete. Two effects have to be considered here. First, the field of view of the irradiance optical inlet differs with distance to the cloud layer. A low stratus is more smoothed than a high cirrus, which is closer to the sensor. Therefore, the variability along a flight leg is mostly dominated by the cirrus inhomogeneities. Between the different legs, the stratus field might have changed and caused the differences of the mean values. Below the cirrus, these differences of the leg averages are in the range of the variability along a leg. In the third cirrus leg, the mean irradiance is increased due to the cirrus. This increase is a range similar to the standard deviation of the three upper legs. This indicates that the variability of the upper three legs is caused by the cirrus and not the stratus.
In the revised manuscript, we added following explanation:

*Assuming that along the flight leg the low stratus is homogeneous with respect to the field of view of the irradiance optical inlet, these higher standard deviations are mainly caused by the spatial variability of the cirrus. The cirrus is located vertically closer to the irradiance sensor and, therefore, smaller horizontally inhomogeneities are resolved by the measurements.*

We added a Satellite picture (Figure 5) where you can see the cirrus/stratus situation.

**Lines 405-410: This is important! – It should be mentioned more clearly in the abstract. Please emphasize that only collocated irradiance measurements of the Learjet and the AIRTOSS give meaningful heating rates. Also, specify which heating rates are theoretically expected instead of only listing the corresponding references.**

In the revised abstract we included this conclusion by:

*"Due to unavoidable biases of the measurements between the individual flight legs, the single platform approach failed to provide a realistic solar heating rate profile while the uncertainties of the tandem approach are reduced. Here, the solar heating rates range up to 6 K day-1 at top of the cirrus layer."*

Literature values of solar heating rates between 0.2-0.5 K/day were reported by Buchholtz et al. (2010) and Thorsen et al. (2013) for subvisible and optically thin cirrus. With an optical thickness of 0.6, the observed cirrus was optically thicker and higher heating rates can be expected. In the revised manuscript we added:

*For subvisible and optically thin cirrus, they calculated heating rates in the range of 0.2-0.5 J day-1. These higher values might result from the higher optical thickness, $\tau=0.6$, of the cirrus observed by AIRTOSS or be caused by horizontal inhomogeneities of the observed cirrus leading to horizontal photon transport as discussed by Finger et al. (2016).*

**Line 407: Here you mention that a cirrus geometrical thickness of more than 200m is too large to allow for positioning of the Learjet above and the AIRTOSS below the cloud layer. Earlier you stated a longer steel wire length – please clarify why the AIRTOSS cannot be positioned below thicker clouds?**

We used a maximum length for the steel wire of 3000 ft (914 m). With this length and a speed of 165 m s$^{-1}$, the AIRTOSS was positioned 180 m below and 896 m behind the aircraft. This caused a temporal misalignment of 5 s. During this campaign, we didn't extend the length of the steel wire rope, because the restricted measurement area would have been to

small to keep the AIRTOSS under control. In addition, we didn't want to increase the temporal misalignment. We added this information to the manuscript.

> *During the AIRTOSS-ICE cam- paign the steel wire was only released to a length of up to 914m (3000 ft) to keep AIRTOSS under manageable conditions within the borders of the the relatively small restricted military areas. Under these conditions and with an airspeed of 165 m s−1 , AIRTOSS stayed approximately 180 m below and 900 m behind the Learjet. This horizontal displacement introduces a delay of about 5 s between Learjet and AIRTOSS instantaneous location.*

**Line 427-428: What exactly can you derive by combining microphysical and radiative measurements. You did show several graphs of collocated measurements but it become not quite clear how this knowledge can be used. – Is it possible to validate radiative transfer retrievals of particle size (based on measured radiative properties) with the simultaneously measured particle size distributions? Or how else can the measurements be used for more in-depth cirrus studies?**

Yes, this was one of the main motivations for why the AIRTOSS was developed. Such a closure study was already published by Finger et al. (2016). In situ cloud microphysics of another cirrus case were used in radiative transfer simulations to calculate the cirrus optical layer properties. At the same time, the collocated irradiance measurements on AIRTOSS were used to derive the optical layer properties and were compared to the model results. This comparison helped to quantify the impact of ice crystal shape, effective radius, and optical thickness on the cirrus radiative forcing. We added the reference to Finger et al. (2016) in the conclusion of the revised manuscript.

> *Further results are presented by Finger et al. (2016) in a closure study, which combines in situ cloud and radiative measurements to quantify the impact of ice crystal shape, effective radius, and optical thickness on cirrus radiative forcing.*

**Line 443-448: Only here you mention that the shown results are taken from a proof-of-concept campaign and that thus the AIRTOSS steel-wire was not extend further. – Please mention that in the very beginning of the manuscript.**

We didn't extend the steel-wire further, because we needed to keep the AIRTOSS at a manageable distance in the relatively small restricted areas. This information is added in the manuscript.

**Section 2.6: The trace gas measurements seem totally unrelated to the paper in which you are focusing on collocated measurements microphysical and radiative properties. Unless you convince me how they add to the entire story, I would suggest to remove the parts referring to the trace gas measurements. You only briefly refer to the trace gas measurements again in lines 455-457. – This is not sufficient to justify the inclusion of the trace gas measurement description.**

As pointed out in your comment, we do not show a case where trace gas data do play a central role since we observed the particles in the upper troposphere. However, specifically at the tropopause the additional information on the tracers (specifically N2O) provides some unique information on the tropopause location to the tandem observations and thus the full setup. Mueller et al. (2015) used these measurements during AIRTOSS-ICE on the Learjet to identify the occurrence of cirrus particles in stratospheric air masses by the amount of N2O, which demonstrate the importance of the full payload for the measurement concept. The N2O instrument was further flown for the first time during AIRTOSS-ICE. We therefore see the trace gases as part of the full technical tandem setup and thus would like to keep this section. Since we would like to publish the manuscript in the AM*Techniques* journal, which is dedicated to publishing advances in remote sensing and in-situ measurement techniques. In our understanding, this also includes the documentation and information about the complete payload of the tandem platform including the trace gas instruments as part of the full measurement concept.

**Minor Comments:**

**Sometimes you refer to the towing sensor shuttle as AIRTOSS, sometimes as the AIRTOSS. Be consistent and choose if you want to call it a noun or if you want to refer to it as proper name.**
Thanks for the comment, we want to use a name for it and changed it in the manuscript.

**Line 4: "detached from" should be extended by "detached from the aircraft via a cable" to illustrate the setup more clearly**
*We changed it.*

**Line 6: replace "layer clouds" by the more scientific term "stratiform clouds"**
*Changed.*

**Line 6: motivate why you need "sophisticated numerical flow simulations" - to quantify shattering effects on the CCP?**
*Changed it to: Sophisticated numerical flow simulations were conducted in order to optimally integrate an axially asymmetric Cloud Combination Probe (CCP) inside AIRTOSS.*

*Line 9-10: move this sentence about the steel cable to line 4 for clarity*

Already changed.

*Line 13 (and 287): The sentence seems backwards: ice crystals grow from small to large sizes (via diffusional growth/aggregation), thus the sentence should be phrased: ...maximum size in the observed...increases from 30mum to 300mum with decreasing altitude.*

We changed it.

*Also, shouldn't the change in maximum size of the PNSD rather refer to geometrical cloud depth than merely altitude? Please clarify.*

We used this explanation to describe the figure. A few sentence later we explain why the cloud particles are distributed like that.

*Line 16: Remove "consequently" or replace it by "thus"*

It is just a synonym. We prefer "consequently".

*Line 16: Add "growth" between microphysical and process*

Changed!

*Line 17: is the solar downward irradiance on the Learjet measured above/in/below the cirrus?*
*Line 18: Clarify where the cloud is positioned with respect to the tandem platform to determine heating rates*

The tandem platform did sample the cirrus at different altitudes. During the profile both platforms had been below, in, and above the cirrus. From the measurements at different altitudes, profiles of heating rates are derived. To clarify this approach in the abstract, we changed this part to:

> *Measurements of solar downward and upward irradi- ances at 670 nm wavelength were conducted above, below, and in the cirrus on both, the Learjet and AIRTOSS. The observed variability of the downward irradiance below the cirrus reflects the horizontal heterogeneity of the observed thin cirrus.*

*Line 25: THEIR microphys. Prop. ; warm or cool (plural!)*

Thank you. We changed it.

*Line 26-28: rearrange sentence structure to proper English. "Especially the ice particle shape was found to determine ... (e.g., Wendisch ... )"*

Changed.

**Line 29: You cannot talk about "such effects" of surface roughness when you haven't previously talked about surface-roughness. – Modify the sentence accordingly.**
Changed.

**Line 47: Clarify if the "two helicopter borne platforms" refer to two helicopters flown simultaneously or if not, what kind of platforms you refer to.**
Changed.

**Line 54: Replace "speed" by "aircraft velocity"**
We changed it.

**Line 55: released by means of a steel wire**
Changed.

**Line 56: In "the study of" Frey et al....**
We changed it.

**Line 58: "this" not "his"**
Thank you.

**Line 60: If the Frey et al. 2009 study is based on the proof-of-concept campaign, it should be mentioned clearly. Also, the proof-of-concept sentence should be moved before line 56. Try to ease the reader into the subject, go from larger picture to more detailed description.**
Changed.

**Line 94: What is the limited distance? Give a value.**
Unfortunately, we are not able to give a precise number for the distance.

**Line 103: Title of this subsection should be "Specifications of the AIRTOSS"**
We changed it.

**Line 113: remove comma**
Thanks.

**Line 121: "of up to 914m"**
Changed.

**Line 128: "less than the maximum ..."**
Changed.

*Line 137: to save energy*
Thanks.

*Line 138: explain abbreviation CCP-CDP*
Okay.

*Line 139: a voltage*
Changed.

*Line 141: no commas*
Changed.

*Line 153: mounted on*
Changed.

*Line 154: Seems like a word is missing after particle-by-particle data analysis/algorithm/technique?*
We made it more clear.

*Line 158: Specify what you mean by size: maximum dimension?*
It is the maximum dimension diameter. We corrected it.

*Line 163: citations should be given in chronological order*
Changed.

*Line 172: Again, this last sentence seems like it was added as an afterthought. Consider moving it after the reference to Knollenberg, maybe by combining those two sentences.*
Changed.

*Line 178: at the bottom*
Thank you.

*Line 180: wavelengths*
Changed.

*Line 180: irradiance sensor; give reference for horizontal alignment requirement*
Changed.

*Line 191: ...symmetric, ... (comma)*
Changed.

*Line 194-197: this sentence needs to be simplified or devided into two for clarity. What do you mean by "aiming at their compensation"?*
We meant: "with the goal to compensate these effects". We changed the sentence though.

**Line 219: As a result, ...**
Changed.

**Line 235: Accordingly, ...**
Thank you.

**Line 272: of less than...**
Thanks.

**Line 293: growth process**
Changed.

**Line 294: water vapor diffusion; the particles don't descent, they sediment**
Changed.

**Line 300: explain the term area ratio**
Regarding to Frey (2011), it is just the area of the shadowed pixels (measured by e.g. the CCP-CIPg instrument) divided by the calculated particle area using the maximum dimension diameter. We added this information in the manuscript.

**Line 304: what orientation was assumed for the falling columnar ice crystal?**
As you can see from the area ratio, the ice crystal is horizontally orientated. To make this more clear, we mentioned it in the manuscript.

**Line 304: replace numbers with "estimated terminal fall velocities"**
Thanks.

**Line 307: Why does aggregation only occur several hours after particle formation at such ice particle number concentration? – Try to present the reader with a good story, instead of with many questions.**
Because the probability for collision is low. We added it.

**Line 326: What do you mean by "undisturbed"? constant?**
Thank you for this comment. We changed the sentence to:

> *Above the cirrus, the downward irradiance is almost constant over the entire legs indicating clear sky for both platforms.*

**Line 349: add citation**
Inserted.

**Line 350: is affected by what? Do you mean "shows variation"?**
Exactly, thank you.

**Line 359: the "in-cloud" inhomogeneities**
Already changed.

**Line 363: Start the sentence with "to make measurements comparable, ..."**
Changed.

**Line 367: Sentence is unclear. Please clarify what the horizontal bars indicate: the standard deviation along individual flight legs or the variability of the radiation along the flight legs?**
Changed.

**Line 406: why radiance? I suppose you mean "irradiance"?**
Correct.

**Line 420: Is SMART really a sensor?**
Changed.

**Line 426: Remove comma**
Thanks.

**Line 454: Again, the reader wonders: What is the higher sampling rate? – Please mention it and relate to the sampling rate and the sample area of the CCP.**
To explain it better we used the sample volume and changed the manuscript.

> *To perform microphysical measurements with a higher temporal resolution, the implementation of holographic instruments is also an attractive alternative. These instruments have a larger sample volume of up to $305 cm^3$, which is much higher than the sample volume of the CCP-CDP instrument ($45 cm^3$ for an aircraft velocity of $165\ m\ s^{-1}$).*

**Questions and Answers regarding RC2:**

***On reading the abstract I was not convinced why I needed to use this system. I think the paper needs to do a bit more to convince the reader that this is a useful technique.***

We made it more clear in the abstract why this system is a useful technique.

> *Vertically resolved solar heating rates were derived by either using single platform measurements in different altitudes or by making use of the collocated irradiance measurements in different altitudes of the tandem platform. Due to unavoidable biases of the measurements between the individual flight legs, the single platform approach failed to provide a realistic solar heating rate profile while the uncertainties of the tandem approach are reduced. Here, the solar heating rates range up to 6 K day−1 at top of the cirrus layer.*

***line115. Does this mean that certification is limited to one payload and any changes require another certification?***

Yes, this is typical for airborne research platforms. The certification process is linked to a specific configuration. Nevertheless, it is possible to certify multiple configurations from the beginning for one platform. Then you are allowed e.g. to change instruments during a campaign.

***section 2.6. It would seem more natural to move this section to just after section 2.3 or 2.4. Or move the flow simulation earlier. At the moment the flow simulation section sits in the middle of sections describing instrumentation.***

Yes, you are right. Thank you for the comment. We moved the flow simulation section to the end of Section 2.

***line 260 0ppb - is it really that sensitive?***

We got that information from the manual. A more detailed look in Köllner (2013) showed that the lower threshold is at 0.9 ppb for 700hPa. We changed it.

***line 295 - do you mean smaller ice crystals nearer the top (lower fallspeeds and hence longer residence times at that altitude) ?***

Yes, exactly. To make it more clear, we changed a few words.

***line 307. What was the relative humidity with respect to ice? Can you reconcile the 2D imagery in figure 6 with the diffusion grown images in Bailey and Hallett 2009 JAS fig5 for your temperature and humidity range?***

We looked into Baily and Hallett, but also into Heymsfield and Miloshevich, JAS, (2005). The particle shape and size look similar. That is the case, because we were under similar conditions (RH ~ 102%, Temp: -35 to -45°C). Unfortunately, the CCP-CIPg instrument does not deliver as good of a resolution like the instruments in the other publications. For that reason, we mentioned in the conclusions that a holographic instrument would be a good

option for future campaigns.

**line 358-360. This is the heart of the reason for flying a tandem formation. If you have one platform within cloud measuring the downwelling radiation and another platform slightly below measuring the same radiation then the difference between those two signals is going to provide information about the intervening cloud. It should not matter that one platform is not at cloud top. Perhaps the errors in the radiation measurements are too large to do this with the separation that was being used? Could a calculation be done to estimate what thickness is required?**

Yes, this is correct. In general, having both platforms in the cloud still provides the cloud properties in the intervening layer. We actually analyzed this when calculating the profile of heating rates in Section 4.2. For the tandem approach, heating rates between both platforms are derived. The results of this exemplary case showed that, in general, the separation was still sufficient to derive cloud optical properties between the two platforms with reasonable uncertainty. However, the current distance used for the measurement setup is at the limit for resolving differences in the irradiance profiles in case of thin cirrus. This is obvious by the large uncertainties estimated for the heating rates in Fig. 8.  Similar conclusions had been made for a second case analyzed by Finger et al. (2016).
In the revised manuscript we removed the original statement and added the following discussion:

> *However, the approach by Werner et al. (2014) for analyzing the collocated number concentration and cloud remote sensing works only if the radiation measurements are performed well above the cloud. In the case of the AIRTOSS-Learjet tandem this would limit the analysis to the uppermost cirrus layer. However, operating radiation measurements on both platforms, the cloud optical layer properties can be derived as presented by Finger et al. (2016). Using the collocation for cloud layers well inside the cloud can also be analyzed.*

**It should now be possible to do a closure study where the microphysical information from AIRTOSS is assumed to represent a column of cloud between AIRTOSS and the Lear. An average along the leg could be used. This column can then be modelled with a radiation code to estimate the effect on the radiation. The radiative response of this column of cloud can then be compared with the measured radiative difference. To me this would be the unique selling point of this system- the ability to carry out this type of analysis. This sort of closure study could be used to try and constrain unobserved quantities such as crystal roughness.**

Yes, this was one of the main motivations for why the AIRTOSS was developed. Such a closure study was already published by Finger et al. (2016). In situ cloud microphysics of another cirrus case were used in radiative transfer simulations to calculate the cirrus optical

layer properties. At the same time the collocated irradiance measurements on AIRTOSS were used to derive the optical layer properties and were compared to the model results. This comparison helped to quantify the impact of ice crystal shape, effective radius, and optical thickness on the cirrus radiative forcing. We added the reference to Finger et al. (2016) in the conclusion of the revised manuscript.

> *Further results are presented by Finger et al. (2016) in a closure study, which combines in situ cloud and radiative measurements to quantify the impact of ice crystal shape, effective radius, and optical thickness on cirrus radiative forcing.*

***Fig8. Yes, this plot is good. The advantage of using the tandem platform for heating rates over single platforms should be emphasized more in the abstract.***

Thank you! We mentioned it in the abstract.

Manuscript prepared for Atmos. Meas. Tech.
with version 2015/11/06 7.99 Copernicus papers of the LATEX class copernicus.cls.
Date: 14 July 2017

**A tandem approach for collocated  measurements of microphysical and radiative cirrus properties**

Marcus Klingebiel[1,2], André Ehrlich[3], Fanny Finger[3], Timo Röschenthaler[4,5],
Suad Jakirlić[5], Matthias Voigt[6], Stefan Müller[4,6], Rolf Maser[4],
Manfred Wendisch[3], Peter Hoor[6], Peter Spichtinger[6], and Stephan Borrmann[2,6]

[1]Max Planck Institute for Meteorology, Atmosphere in the Earth System Department, Hamburg, Germany
[2]Max Planck Institute for Chemistry, Particle Chemistry Department, Mainz, Germany
[3]Leipzig Institute for Meteorology (LIM), University of Leipzig, Leipzig, Germany
[4]Enviscope GmbH, Frankfurt, Germany
[5]Institute for Fluid Mechanics and Aerodynamics, Darmstadt University of Technology, Darmstadt, Germany
[6]Institute for Atmospheric Physics, Johannes Gutenberg University Mainz, Mainz, Germany

*Correspondence to:* S. Borrmann (stephan.borrmann@mpic.de)

**Abstract.** Microphysical and radiation measurements were collected with the  novel AIRcraft TOwed Sensor Shuttle (AIRTOSS) - Learjet tandem platform.  The platform is a combination of  an instrumented Learjet 35A research aircraft and an  aerodynamic bird, which is detached from and retracted back to the aircraft during flight via a steel wire with a length of 4000 m. Both platforms are equipped with radiative, cloud microphysical, trace gas  and meteorological instruments. The purpose of the development of this tandem setup is to study the inhomogeneity of cirrus as well as other  stratiform clouds. Sophisticated numerical flow simulations were conducted in  order to optimally integrate  an axially asymmetric Cloud Combination Probe (CCP) inside  AIRTOSS. The tandem platform was  applied during measurements at altitudes up to 36 000 ft (10 970 m)  in the framework of the AIRTOSS - Inhomogeneous Cirrus Experiment (AIRTOSS-ICE).  Ten flights were performed above the North Sea and Baltic Sea to probe frontal  and in-situ formed cirrus,  as well as anvil outflow cirrus.  For one flight, cirrus microphysical and radiative properties displayed significant inhomogeneities resolved by both measurement platforms.  The CCP data show that the maximum of the observed particle number size distributions shifts with  decreasing altitude from 30 μm to 300 μm, which is typical for frontal, midlatitude cirrus. Theoretical considerations imply that cloud particle aggregation inside the studied cirrus is very unlikely. Consequently, diffusional growth was identified to be the dominant microphysical growth process. Measurements of

solar downward  and upward irradiances at 670 nm wavelength  were conducted above, below, and in the cirrus on both, the Learjet and  AIRTOSS. The observed variability of the downward irradiance below the cirrus reflects the horizontal heterogeneity of the observed thin cirrus.  Vertically resolved solar heating rates were derived by either using single platform measurements in different altitudes or by making use of the collocated irradiance measurements in different altitudes of the tandem platform,  . Due to unavoidable biases of the measurements between the individual flight legs, the single platform approach failed to provide a realistic solar heating rate profile while the uncertainties of the tandem approach are reduced. Here, the solar heating rates range up to $6\,\mathrm{K\,day^{-1}}$  at top of the cirrus layer.

**1  Introduction**

Cirrus clouds consist of ice particles and occur in the upper troposphere and lower stratosphere at temperatures below $-38\,^{\circ}\mathrm{C}$ (Boucher et al., 2014; Koop et al., 2000).  Their wide range of microphysical and macrophysical properties  affects the solar and terrestrial radiative budget of the  Earth's climate system. Depending on the microphysical properties  cirrus either warms or cools the layer below the clouds (Lynch, 2002; Zhang et al., 1999).  Among other factors, the ice particle shape  determines the cirrus radiative properties such as  its albedo or spectral radiative layer properties  (e.g., Wendisch et al. (2005), Wendisch et al. (2007), Eichler et al. (2009) or Finger et al. (2016)). Ice particle shape and surface roughness may  also cause biases in retrievals of cirrus properties from satellite measurements.

 To quantify the dependence of the cloud radiative forcing  from cloud properties,  vertically separated observations of the cirrus microphysical and radiative properties are needed. This can be realized by consecutive measurements by one single  aircraft or collocated observations by two platforms. The first approach is  problematic due to the (usually too large) temporal  displacement between the observations in, below, and above the cloud. Collocated measurements using two coordinated aircraft were attempted for example during the Cirrus Regional Study of Tropical Anvils and Cirrus Layers - Florida Area Cirrus Experiment (CRYSTAL-FACE) in 2002 (Jensen et al., 2004), the Tropical Composition, Cloud and Climate Coupling (TC4) mission in 2007 (Toon, 2007), and the Radiation-Aerosol-Cloud Experiment in the Arctic Circle (RACEPAC) in 2014 (Ehrlich and

Wendisch, 2015). However, as pointed out by Frey et al. (2009) and others,  the exact vertical collocation between the two aircraft with different speeds is problematic as well. To minimize these  collocation issues, towed sensor systems have been applied  in the past.

During the CARRIBA (Cloud, Aerosol, Radiation and tuRbulence in the trade wInd regime over BArbados) project (Siebert et al., 2013) two  platforms connected by a cable to a helicopter were applied to obtain collocated measurements of thermodynamic, turbulent, microphysical, and radiative properties within clouds. Werner et al. (2014) showed that such observations can be used to link cloud microphysical and radiative properties and estimate the Twomey effect in shallow cumulus. However,  such helicopter measurements are limited to altitudes below 3000 m and, therefore, are not suited for investigating cirrus.

Frey et al. (2009) introduced a new tandem measurement platform consisting of a Learjet 35A research aircraft and an AIRcraft TOwed Sensor Shuttle (AIRTOSS), which can operate in higher altitudes and  velocities (~ 700 km h$^{-1}$). AIRTOSS is a sensor pod that is attached under the right wing of the Learjet. When the Learjet reaches the measurement area, AIRTOSS is released and towed by the aircraft  via a steel wire. In 2007 a proof-of-concept campaign was conducted to evaluate the technical feasibility, the flight safety, and the flight performance of AIRTOSS. In the study of Frey et al. (2009), AIRTOSS was  equipped with a Cloud Imaging Probe (CIP) to measure the microphysical properties of the clouds and two navigation systems . At this time, the configuration of the tandem platform was certified only to fly up to an altitude of 25 000 ft (7620 m), which is below the altitude where most cirrus typically occurs.  Frey et al. (2009) show that turbulence as well as acceleration and deceleration maneuvers should be avoided to keep roll and pitch angles in a range of $\pm$ 3 $^\circ$  which appears tolerable for irradiance measurements (by definition related to a strictly horizontal receiving plane).

In this paper an advanced AIRTOSS platform including radiative and cloud microphysical instruments  is introduced, which is certified for higher altitudes up to 41 000 ft (12 500 m).  Technical details of the redesigned  AIRTOSS are presented in Section 2. Section 3 shows  results of collocated measurements in cirrus clouds with the Learjet 35A and

[revised manuscript text omitted]

190 In comparison to the CCP-CIPg instrument, the CCP-CDP detects particles in a smaller particle diameter size range between 2 μm and 50 μm. The instrument is based on forward light-scattering with a light collection angle from 4 ° up to 12 ° and uses a laser diode with a wavelength of 658 nm. A sample area of $0.27 \pm 0.025$ mm$^2$ was estimated by using a piezoelectric droplet generator laboratory setup, similar to the design of  Wendisch et al. (1996) and
195 Lance et al. (2010). The accuracy and prior measurements of the CCP-CDP instrument are shown in

Molleker et al. (2014) and Klingebiel et al. (2015).

The Learjet was equipped with a Forward Scattering Spectrometer Probe (FSSP) inside the sensor pod (Figure 1b). This instrument was developed by Knollenberg (1976) to measure particles in a size range between 2 μm and 47 μm diameter and is a predecessor of the CCP-CDP (Brenguier et al., 2013). Because the FSSP has neither mounted tips nor the feasibility to exclude shattered particles by software algorithms, here it was mainly used for testing purposes and as a cloud indicator during the campaign. In the future it will be replaced with more advanced instrumentation.

**2.4 Spectral solar radiation measurements**

To measure the up- and downward irradiance of a cirrus layer located between the Learjet and  AIRTOSS, both platforms were equipped with the Spectral Modular Airborne Radiation measurement sysTem (SMART). For each radiation component (upward/downward irradiance), SMART combines two Zeiss Spectrometers each connected by fibre wires to an optical inlet mounted on the top or at  the bottom of AIRTOSS and the Learjet. The spectral range of SMART is between 300 nm and 2200 nm with a resolution of 3 nm for  wavelengths below 1000 nm and 9 – 16 nm above (Wendisch et al., 2001; Bierwirth et al., 2009). The upward looking  irradiance sensor on the Learjet was placed on a stabilized platform to keep it horizontally aligned during the flights.

Due to the limited space inside AIRTOSS (see Figure 2a), an active  horizontal stabilization of the radiation sensors could not be realized. For this reason an Inertial Navigation System (INS) in combination with a Global Positioning System (GPS) was used to record attitude and alignment angles. This data was screened afterwards to identify and remove sections where reliable measurements were not possible. A detailed analysis of the solar radiation instruments, the measurements in cirrus and the scientific results of the AIRTOSS-ICE campaign are given in Finger et al. (2016).

**2.5  Trace gas instruments**

Besides the radiation and microphysical instruments, the AIRTOSS-Learjet tandem platform was equipped with a suite of instruments quantifying the concentration of different trace gases.

The Fast Aircraft-Borne Licor Experiment (FABLE) was integrated on the Learjet to detect the amount of carbon dioxide ($CO_2$) at flight altitude (Gurk et al., 2000). Nitrous oxide ($N_2O$) and carbon monoxide (CO) were measured with the University of Mainz Airborne QCL-Spectrometer (UMAQS, see Mueller et al. (2015) for details).

Temperature and relative humidity measurements were made on the Learjet and on AIRTOSS by the MOZAIC Capacitive Hygrometer (MCH) which belongs to the Measurement of OZone by AIRBUS In-Service AirCrafts (MOZAIC) system. The MCH uses a capacitive sensor and a Pt100

element to measure the relative humidity and the temperature respectively. The accuracy is $\pm 0.5\,°C$ for the temperature measurement and $\pm 5\,\%$ for the detection of the relative humidity. Evaluation- and measurement-methods of the MCH are described in detail in Neis et al. (2015).

Water vapor measurements were taken by the Fast In-Situ Hygrometer instrument (FISH) and the Selective Extractive Airborne Laser Diode Hygrometer II (SEALDH-II). The FISH instrument is developed and operated by the *Forschungszentrum Jülich*. It is based on Lyman-Alpha-Photometry and detects water vapor in a range between 1 ppmv and 1000 ppmv with an uncertainty of $\pm$ 0.2 ppmv (Zöger et al., 1999). SEALDH-II is operated by the *Physikalisch-Technischen Bundesanstalt*, uses direct Tunable Diode Laser Absorption (dTDLAS) and leads without any previous gas-based instrument calibration to an absolute $H_2O$ concentration value. It operates in a detection range between about 30 ppmv and roughly 40 000 ppmv with an accuracy of 0.35 % and a time resolution of $<1\,s$ (Buchholz et al., 2016; Buchholz and Eb

Ozone ($O_3$) measurements were performed on the Learjet by using a UV-Photometry 42 M Ozone Analyzer developed by *Environment S.A.*. This instrument detects the UV-absorption caused by $O_3$ at a wavelength of 254 nm in a measurement range between 0.9 ppb (at 700 hPa) and 10 000 ppb with an uncertainty of 10 % (Köllner, 2013). These instruments can be used for independent trace gas dynamics studies (e.g. Mueller et al. (2015)), for better finding the exact location of the tropopause, identifying tropopause folds, as well as stratospheric influence on uppermost tropospheric cirrus clouds (especially subvisual cirrus), finding borders of air masses (e.g. the polar dome), among others.

[revised manuscript text omitted]

365 of the shadowed pixels (detected by the CCP-CIPg) divided by the calculated particle area using the maximum dimension diameter (Frey, 2011). As an example, a spherical (area ratio = 1) and a horizontal orientated column shaped (area ratio = 0.25) ice particle with an initialized diameter of $D_p = 200$ µm are assumed. This represents the measured conditions during Flight Leg 3 at an altitude of 9333 m (see Figure 7). For the spherical particle, a terminal velocity of $v_t = 91\,\mathrm{cm\,s}^{-1}$ was calcu-

370 lated, while for  the horizontal orientated columnar particle $v_t = 14.5\,\mathrm{cm\,s}^{-1}$ was estimated. With these  estimated terminal fall velocities, the particles would need 11 minutes and 71 minutes, respectively, until they reach the bottom layer of the cloud at an altitude of 8716 m.

Following the discussion by Kienast-Sjögren et al. (2013), particles with a number concentration of $5.8 \cdot 10^{-2}$ cm$^{-3}$ (Level 3 in Figure 7) need at least several hours before aggregation processes occur, because the probability for collision is low. For this reason, aggregation is unlikely, and diffusional growth seems to be the dominant process for this particular cirrus observed during AIRTOSS-ICE.

**3.2 Solar downward irradiance**

In addition to the microphysical measurements, collocated measurements of spectral solar radiation were performed during the cirrus event of Section 3.1. Similar to Figure 6a, a profile of the spectral downward irradiance (at 670 nm wavelength) measured by SMART on AIRTOSS and Learjet is given in Figure 6b. The individual legs were filtered for turns of both platforms which assures that only level flight conditions were considered. Additionally, only legs flown in the same direction and above the same locations were chosen to assure similar cloud and surface conditions below the cirrus. In total, five legs with simultaneous measurements on AIRTOSS and the Learjet are available with larger vertical separation in the cirrus and less separation at cloud top and above. The impact of the cirrus on the downward irradiance is most obvious in the two lower legs where the radiation is attenuated by the cirrus. The attenuation is highly variable due to the horizontal heterogeneity of the cirrus. However, both sensors on AIRTOSS and Learjet show almost the same pattern, illustrating the collocation of the measurements. The similarity in the two datasets also results from the small vertical displacement of Learjet and AIRTOSS of less than 200 m. During the higher flight legs, the attenuation of downward irradiance by the cirrus is significantly lower. In the third leg, only AIRTOSS measurements are slightly affected by the cirrus, while the Learjet already observed clear sky conditions. Above the cirrus, the downward irradiance is almost  constant over the entire legs indicating clear sky for both platforms.

**4 Discussion**

Two cases are selected to illustrate the potential of the collocation of measurements achieved by the AIRTOSS-Learjet tandem platform. Due to the different instruments operated on AIRTOSS and Learjet different combined analysis of data are possible. Beside combining in-situ and radiation measurements also the simultaneous radiation measurements on both platforms can be analyzed jointly.

**4.1 Collocation of microphysical and radiative properties**

Figure 8 shows a time series of downward spectral irradiance at 670 nm wavelength measured from the Learjet (Figure 8a) and AIRTOSS (Figure 8b)  during a flight leg observed on  4 September 2013 between 09:35 UTC and 09:39 UTC, when AIRTOSS was operated at an altitude

of around 9900 m. In addition, Figure 8c shows the detected number concentration of the CCP-CDP and the CCP-CIPg.  The cloud particle number concentrations above zero were detected within two sections of the flight leg and  indicate that AIRTOSS did penetrate two cirrus filaments at the top of the cirrus layer. The downward irradiance has been constant for most of the flight leg indicating clear sky conditions without attenuation of the incoming solar radiation. The strongest deviation from the clear sky conditions was found at about 09:38:05 UTC where the irradiance shows a rapid decrease for both platforms. This coincides with higher values in the particle number concentration measurements. The increasing number concentration indicates that AIRTOSS is located in a thicker part of the sampled cloud and certainly the cloud top is above AIRTOSS. As the Learjet measurements are located closer to cloud top the effect is here smaller compared to the AIRTOSS observations. At cloud edges also an increase of the irradiance can occur due to three-dimensional radiative effects (Sabburg and Long, 2004). For the smaller cloud observed at the beginning of the leg (09:35:45 - 09:36:40 UTC), only the downward irradiance measured by AIRTOSS shows variation, while the downward irradiance measured by the instruments on the Learjet remains almost constant. At this time only AIRTOSS was located inside the cirrus while the Learjet flew above cloud top and consequently only the downward radiation in the altitude of AIRTOSS was reduced.

Such constellations are well suited to investigate the interaction of cloud microphysical and radiative properties as demonstrated by Werner et al. (2014) for shallow cumulus. However,  the approach by Werner et al. (2014) for analyzing the collocated number concentration and  cloud remote sensing works only if the radiation measurements are performed well above the cloud. In the case of the AIRTOSS-Learjet tandem this would limit the analysis to the uppermost cirrus layer. However, operating radiation measurements on both platforms, the cloud optical layer properties can be derived as presented by Finger et al. (2016). Using the collocation for cloud layers well inside the cloud can also be analyzed.

**4.2 Vertical profile of solar heating rates**

[revised manuscript text omitted]

This improvement in calculating heating rates illustrates the benefit of collocated irradiance measurements. However, the derived heating rates still do not represent theoretical results as provided by e.g., Bucholtz et al. (2010) and Thorsen et al. (2013).  For subvisible and optically thin cirrus, they calculated heating rates in the range of 0.2 - 0.5 K day$^{-1}$. These higher values might result from the higher optical thickness, $\tau = 0.6$, of the cirrus observed by AIRTOSS or be caused by horizontal inhomogeneities of the observed cirrus leading to horizontal photon transport as discussed by Finger et al. (2016).

**5 Conclusions**

The advanced AIRTOSS-Learjet tandem platform was applied during the AIRTOSS-ICE campaign  to perform collocated measurements of cirrus cloud properties. A combination of the Learjet and  AIRTOSS, both equipped with  radiation and microphysical in-situ instruments, allowed for measurements of  cirrus properties in different altitudes using just one aircraft. The new certification for the AIRTOSS-Learjet tandem platform  enabled to probe cirrus  at altitudes up to 36 000 ft (10 970 m). The campaign  showed that collocated measurements with the   revised AIRTOSS-Learjet tandem platform are feasible. This

is demonstrated by combining the microphysical and radiative measurements and, as an illustrative example, by deriving solar heating rates. Further results are presented by Finger et al. (2016) in a closure study, which combines in situ cloud and radiative measurements to quantify the impact of ice crystal shape, effective radius, and optical thickness on cirrus radiative forcing.

515  A case study is presented where AIRTOSS-ICE measurements  are used to derive vertical profiles of cloud microphysical and radiative properties. Using the profiles of upward and downward irradiances, it is shown that solar heating rates can be estimated with  an improved accuracy when collocated measurements are applied, instead of using a single platform. Despite the  expected higher uncertainties introduced

520 by the measurement errors from two independent measurement systems, the collocated observations resulted in a more realistic profile of solar heating rates as these are not affected by changes of the radiation field below the  observational altitude (e.g., inhomogeneous surface albedo, lower cloud layers). Observations performed with a single aircraft strongly depend on stable conditions between consecutive flight legs and, therefore, are subject to serious uncertainties in derived

525 profiles of solar heating rates.

However, AIRTOSS-ICE also showed the limits of the collocated measurement setup. The investigated cirrus had a thickness of more than 200 m, which is larger than the distance between Learjet and AIRTOSS during the conducted measurement example. This did not allow for the radiative instruments to measure concurrently with  AIRTOSS below and with the Learjet above the cirrus

530 layer, which would have been needed to derive the cirrus radiative layer properties (Finger et al., 2016). The short distance between both platforms resulted in only small differences in the upward and downward irradiances measured on  AIRTOSS and the Learjet for this  sampling example. An increase of the vertical distance beyond 200 m is  not easy to

535 achieve. It would require a longer steel wire and/or a slower aircraft, as well larger areas where such flights are permitted. For clouds with a larger vertical extent, two single aircraft could be a better choice. It certainly depends on the scientific goals and instrumentation whether or not the AIRTOSS-Learjet tandem platform is the appropriate choice.

With respect to microphysical inhomogeneities, the vertical separation of 200 m between both plat-

540 forms is sufficient for cirrus studies. What would be required additionally are microphysical in-situ instruments with overlapping measurement characteristics, or, ideally, two identical instrument sets on both platforms. To perform microphysical measurements with a higher temporal resolution, the implementation of holographic instruments is also an attractive alternative. These instruments have a larger sample  volume of up to 305 cm$^2$

545  (Schlenczek et al., 2016), which is much higher than the sample volume of the CCP-CDP (45 cm$^3$ for an aircraft velocity of 165 m s$^{-1}$). Furthermore, the integration of trace gas instruments  inside AIRTOSS and the Learjet could be used, e.g., for collocated trace gas

measurements in the vicinity of the tropopause layer, the edges of tropopause folds, streamers etc. To study different atmospheric conditions or to obtain better statistics of cirrus cloud, the operation

550 of the AIRTOSS-Learjet tandem platform in other regions, outside of military restricted areas,  remains a significant challenge. This could be accomplished in less populated  areas, such as the polar regions, remote areas of the oceans, rain forests and others.

*Acknowledgements.* The AIRTOSS-ICE project was supported by the *Deutsche Forschungsgemeinschaft* (DFG) through projects "WE 1900/19-1, BO 1829/7-1, SP 1163/3-1" and on a significant level by internal funds of the

555 *Particle Chemistry Department at the Max Planck Institute for Chemistry*. We particularly thank the pilots and the crew of the *Gesellschaft für Flugzieldarstellung* for making this project possible. We are also thankful for the support of the electrical engineers Wilhelm Schneider and Christian von Glahn (University of Mainz) and all other participants of the AIRTOSS-ICE campaign.

[revised manuscript text omitted]

**Figure 8.** Downward spectral irradiance at 670 nm measured from the Learjet (a) and the AIRTOSS (b) and number concentration (NC) measured on the AIRTOSS platform with the CCP-CDP ($2 - 50\,\mu m$) and the CCP-CIPg ($15 - 960\,\mu m$) instrument (c). The data was obtained at the highest flight leg, measured on 4 September 2013, where the AIRTOSS flow at an altitude of around 9900 m. The vertical bars indicate the error of the instruments and the running average uses the boxcar smoothing algorithm with 10 repetitions.

[Figure]

**Figure 9.** a) Profiles of vertical upward and downward broadband irradiance measured on AIRTOSS and the Learjet. The bars indicate the standard deviation of the irradiance along the individual flight legs. b) Solar heating rates calculated from the irradiance profile either using a single platform or the collocated measurements. The gray area indicates the cirrus layer as indicated by the CCP.